# Deep Recurrent Gaussian Process with Variational Sparse Spectrum Approximation

## Abstract

Modeling sequential data has become more and more important in practice. Some applications are autonomous driving, virtual sensors and weather forecasting. To model such systems, so called recurrent models are frequently used. In this paper we introduce several new Deep Recurrent Gaussian Process (DRGP) models based on the Sparse Spectrum Gaussian Process (SSGP) and the improved version, called Variational Sparse Spectrum Gaussian Process (VSSGP). We follow the recurrent structure given by an existing DRGP based on a specific variational sparse Nyström approximation, the recurrent Gaussian Process (RGP). Similar to previous work, we also variationally integrate out the input-space and hence can propagate uncertainty through the Gaussian Process (GP) layers. Our approach can deal with a larger class of covariance functions than the RGP, because its spectral nature allows variational integration in all stationary cases. Furthermore, we combine the (Variational) Sparse Spectrum ((V)SS) approximations with a well known inducing-input regularization framework. For the DRGP extension of these combined approximations and the simple (V)SS approximations an optimal variational distribution exists. We improve over current state of the art methods in prediction accuracy for experimental data-sets used for their evaluation and introduce a new data-set for engine control, named Emission.

## 1 Introduction

Modeling sequential data for simulation tasks in the context of machine learning is hard for several reasons. Their internal structure poses the problem of modeling short term behavior and long term behavior together for different types of data variables, where the data variables themselves might differ in the information gain in the chosen time frequency. Recurrent models (Hochreiter & Schmidhuber, 1997; Nelles, 2013; Pascanu et al., 2013) have proven to perform well on these tasks. They consist of output-data and input-data structured sequentially for shifted discrete time steps. The general form of a *recurrent model* is given by

$$\mathbf{h}_i = \zeta(\mathbf{h}_{i-1}, \ldots, \mathbf{h}_{i-H}, \mathbf{x}_{i-1}, \ldots, \mathbf{x}_{i-H}) + \boldsymbol{\epsilon}_i^{\mathbf{h}}, \tag{1}$$

$$\mathbf{y}_i = \psi(\mathbf{h}_i, \ldots, \mathbf{h}_{i-H}) + \boldsymbol{\epsilon}_i^{\mathbf{y}}, \tag{2}$$

where $\mathbf{x}_i$ is an external input, $\mathbf{y}_i$ is an output observation, $\mathbf{h}_i$ is a latent hidden representation or state at time $i = H + 1, \ldots, N$, where $N \in \mathbb{N}$ is the number of data samples, $H \in \mathbb{N}$ is the chosen time horizon, $\zeta, \psi$ are non-linear functions modeling *transition* and *observation* and $\boldsymbol{\epsilon}_i^{\mathbf{h}}, \boldsymbol{\epsilon}_i^{\mathbf{y}}$ are transition and observation noise, which are adjusted for the specific problem (details on dimensions and ranges will be specified in upcoming sections).

In control and dynamical system identification previous work on Bayesian recurrent approaches for modeling sequential data usually makes use of *(non-)linear auto-regressive with exogenous inputs models* ((N)ARX) and *state-space models* (SSM), for both see Nelles (2013). The general recurrent model given in Equation (1) and (2) represents both cases. This can be recognized by its general recurrent and hierarchical structure. This work deals with deep learning in a recurrent fashion for modeling sequential data in a Bayesian non-parametric approach by using GPs. To make a connection to the general recurrent model, the deep structure arises by defining $\zeta$ in Equation (1) in a deep manner (Pascanu et al., 2013, Section 3).

To achieve scalability, GPs normally make use of sparse approximations for the covariance function.

This paper proposes DRGP models based on (V)SS approximations, denoted by DRGP-(V)SS. For reproducibility of the experimental results, we provide the code online[1]. Therefore, we follow the same deep recurrent structure as introduced in Mattos et al. (2016). To summarize, the contributions of this paper are the following:

- Extension of the sparse GP based on the SS approximation introduced by Lázaro-Gredilla et al. (2010) and the improved VSS approximation by Gal & Turner (2015) to DRGPs;
- Improvement of regularization properties of the variational bounds through the combination of the (V)SS approximations with the inducing-point (IP) regularization of Titsias & Lawrence (2010);
- Propagation of uncertainty through the hidden layers of our DGPs by variationally integrating out the input-space;
- The existence of an optimal variational distribution in the sense of a functional local optimum of the variational bounds of our DRGPs models is established.

The DRGP of Mattos et al. (2016) is limited to a small class of deterministic covariance functions, because the covariance functions variational expectation has to be analytically tractable. Using the (V)SS approximations instead, we can derive a valid approximation for every stationary covariance function, because the basis functions expectation is always tractable. We show that this approach improves over state of the art approaches in prediction accuracy on several cases of the experimental data-sets used in Mattos et al. (2016); Svensson et al. (2016); Al-Shedivat et al. (2017); Salimbeni & Deisenroth (2017); Dörr et al. (2018) in a simulation setting. For scalability, *Distributed Variational Inference* (DVI) (Gal et al., 2014) is recommended and can lower the complexity from $\mathcal{O}(NM^2 Q_{\max}(L+1))$ down to $\mathcal{O}(M^3)$ for $N \leq M$, $M$ the sparsity parameter, $(L+1)$ the amount of GPs and $Q_{\max}$ is the maximum over all input dimensions used in our defined deep structure for $\zeta$ and $\psi$. Therefore, the number of cores must scale suitably with the number of training-data.

## 2    RELATED WORK TO GAUSSIAN PROCESSES WITH SSM AND DGPS

An *SSM with GPs* (GP-SSM) for transition and observation functions is used by Wang et al. (2005), where the uncertainty in the latent states is not accounted for, which can lead to overconfidence. Turner et al. (2010) solved this problem, but they have complicated approximate training and inference stages and the model is hard to scale. Frigola et al. (2014) used a GP for transition, while the observation is parametric. Svensson et al. (2016) used an approximation of the spectral representation by Bochner's theorem in a particular form and with a reduced rank structure for the transition function. They realize inference in a fully Bayesian approach over the amplitudes and the noise parameters. The construction of Eleftheriadis et al. (2017) involves a variational posterior that follows the same Markov properties as the true state with rich representational capacity and which has a simple, linear, time-varying structure. Dörr et al. (2018) introduced a GP-SSM with scalable training based on doubly stochastic variational inference for robust training. Our models extend the GP-SSM framework by defining the transition as a DRGP based on our newly derived (V)SS approximations in the Sections 3.3, 3.4, where the latent (non-observed) output-data is learned as a hidden state. We refer to the report Föll et al. (2017)[2] for a detailed but preliminary formulation of our models and experiments.

Following Damianou & Lawrence (2013), a *Deep Gaussian Process* (DGP) is a model assuming

$$\mathbf{y}_i = \mathrm{f}^{(L+1)}\big(\mathrm{f}^{(L)}\big(\mathrm{f}^{(L-1)}(\ldots(\mathrm{f}^{(1)}(\mathbf{x}_i) + \boldsymbol{\epsilon}_i^{\mathbf{h}^{(1)}})\ldots) + \boldsymbol{\epsilon}_i^{\mathbf{h}^{(L-1)}}\big) + \boldsymbol{\epsilon}_i^{\mathbf{h}^{(L)}}\big) + \boldsymbol{\epsilon}_i^{\mathbf{y}},$$

where the index $i = 1, \ldots, N$ is *not* necessarily the time and where we define $\mathbf{h}_i^{(1)} \overset{\text{def}}{=} \mathrm{f}^{(1)}(\mathbf{x}_i) + \boldsymbol{\epsilon}_i^{\mathbf{h}^{(1)}}$, $\mathbf{h}_i^{(l+1)} \overset{\text{def}}{=} \mathrm{f}^{(l)}(\mathbf{h}_i^{(l)}) + \boldsymbol{\epsilon}_i^{\mathbf{h}^{(l)}}$, for $l = 2 \ldots, L - 1$, where $L \in \mathbb{N}$ is the number of hidden layers. The noise $\boldsymbol{\epsilon}_i^{\mathbf{h}^{(l)}}$, $\boldsymbol{\epsilon}_i^{\mathbf{y}}$ is assumed Gaussian and the functions $\mathrm{f}^{(l)}$ are modeled with GPs for $l = 1, \ldots, L + 1$. To obtain computational tractability, in most cases variational approximation and inference is used. Damianou & Lawrence (2013) introduced these kind of DGPs based on the sparse variational approximation following Titsias (2009); Titsias & Lawrence (2010). Based on this,

---

[1] DRGP-(V)SS code available from `http://github.com/RomanFoell/DRGP-VSS`.
[2] Available on the website `https://arxiv.org/`.

Dai et al. (2016) introduced a DGP with a variationally auto-encoded inference mechanism and which scales on larger data-sets. Cutajar et al. (2016) introduced a DGP for the so called *Random Fourier Features* (RFF) approach (Rahimi & Recht, 2008), where the variational weights for each GPs are optimized along with the hyperparameters. This approach does not variationally integrate out the latent inputs to carry through the uncertainty and no existence of an optimal variational distribution for the weights is proven to reduce the amount of parameters to optimize in training. Furthermore, Salimbeni & Deisenroth (2017) introduced an approximation framework for DGPs, which is similar to the single GP of Hensman & Lawrence (2014), but does not force independence between the GP layers and which scales to billions of data.

Two state of the art approaches for DRGPs have been introduced by Mattos et al. (2016), the RGP, which we call DRGP-Nyström, based on the variational sparse Nyström/inducing-point approximation introduced by Titsias (2009); Titsias & Lawrence (2010), as well as Al-Shedivat et al. (2017), which we call GP-LSTM, based on deep kernels via a *Long-short term memory* (LSTM) network (Hochreiter & Schmidhuber, 1997), a special type of *Recurrent Neural Network* (RNN).

DRGP-Nyström uses a recurrent construction, where the auto-regressive structure is not realized directly with the observed output-data, but with the GPs latent output-data and uses a Variational Inference (VI) framework, named *Recurrent Variational Bayes* (REVARB). The structure acts like a standard RNN, where every parametric layer is a GP. So additionally uncertainty information can be carried through the hidden layers.

GP-LSTM is a combination of GPs and LSTMs. LSTMs have proven to perform well on modeling sequential data. LSTMs try to overcome vanishing gradients by placing a memory cell into each hidden unit. GP-LSTM uses special update rules for the hidden representations and the hyperparameters through a semi-stochastic optimization scheme. It combines a GP with the advantages of LSTMs by defining structured recurrent deep covariance functions, also called deep kernels, which reduces the time and memory complexities of the linear algebraic operations (Wilson et al., 2016).

# 3    GAUSSIAN PROCESSES AND VARIATIONAL SPARSE SPECTRUM GP

Loosely speaking, a GP can be seen as a Gaussian distribution over functions. We will first introduce GPs and GP regression and then recall the SSGP by (Lázaro-Gredilla et al., 2010) and its improved version VSSGP by (Gal & Turner, 2015). Based on these, we derive new variational approximations. We use the notation $a$, $f_{\mathbf{x}}$, $y$, (italic) for random variables, $\mathbf{a}$, $f(\mathbf{x})$, $\mathbf{y}$ (upright) for realizations/samples and data.

## 3.1    GAUSSIAN PROCESSES

A stochastic process $f$ is a GP if and only if any finite collection of random variables $f_{\mathbf{X}} \stackrel{\text{def}}{=} [f_{\mathbf{x}_1}, \ldots, f_{\mathbf{x}_N}]^T$ forms a Gaussian random vector (Rasmussen, 2006). A GP is completely defined by its mean function $m : \mathbb{R}^Q \to \mathbb{R}, \mathbf{x} \mapsto m(\mathbf{x})$, $Q \in \mathbb{N}$ the input-dimension, and covariance function $k : \mathbb{R}^Q \times \mathbb{R}^Q \to \mathbb{R}, (\mathbf{x}, \mathbf{x}') \mapsto k(\mathbf{x}, \mathbf{x}')$ (Kallenberg, 2006, Lemma 11.1), where

$$m(\mathbf{x}) \stackrel{\text{def}}{=} \mathbf{E}\left[f_{\mathbf{x}}\right], \quad k(\mathbf{x}, \mathbf{x}') \stackrel{\text{def}}{=} \text{cov}(f_{\mathbf{x}}, f_{\mathbf{x}'}) = \mathbf{E}\left[(f_{\mathbf{x}} - m(\mathbf{x}))(f_{\mathbf{x}'} - m(\mathbf{x}'))\right],$$

and the GP will be written as $f \sim \mathcal{GP}(m, k)$. Be aware of that a valid covariance function must produce a positive definite matrix $K_{NN} \stackrel{\text{def}}{=} (k(\mathbf{x}_i, \mathbf{x}_j))_{i,j=1}^N \in \mathbb{R}^{N \times N}$, when filling in combinations of data-input points $\mathbf{x}_i$, $i = 1, \ldots, N$.

Let $\mathbf{y} \stackrel{\text{def}}{=} [\mathrm{y}_1, \ldots, \mathrm{y}_N]^T \in \mathbb{R}^N$, $\mathbf{X} \stackrel{\text{def}}{=} [\mathbf{x}_1, \ldots, \mathbf{x}_N]^T \in \mathbb{R}^{N \times Q}$ and we assume $\mathrm{y}_i = \mathrm{f}(\mathbf{x}_i) + \epsilon_i^{\mathrm{y}}$, where $\epsilon_i^{\mathrm{y}} \sim \mathcal{N}(0, \sigma_{\text{noise}}^2)$, for $i = 1, \ldots, N$, and our aim is to model any set of function values $\mathbf{f} \stackrel{\text{def}}{=} [\mathrm{f}(\mathbf{x}_1), \ldots, \mathrm{f}(\mathbf{x}_N)]^T \in \mathbb{R}^N$ at $\mathbf{X}$ as samples from a random vector $f_{\mathbf{X}}$. Moreover, we assume the prior $f_{\mathbf{X}} | \mathbf{X} \sim \mathcal{N}(\mathbf{0}, K_{NN})$, meaning that any set of function values $\mathbf{f}$ given $\mathbf{X}$ are jointly Gaussian distributed with mean $\mathbf{0} \in \mathbb{R}^N$ and a covariance matrix $K_{NN}$.

The predictive distribution $p_{f_{\mathbf{x}^*} | \boldsymbol{x}^*, \boldsymbol{X}, \boldsymbol{y}}$ for a test point $\mathbf{x}^* \in \mathbb{R}^Q$, where $K_{N*} \stackrel{\text{def}}{=} (k(\mathbf{x}_i, \mathbf{x}^*))_{i=1}^N \in \mathbb{R}^N$, and analogously $K_{**}$, $K_{*N}$, can be derived through the joint probability model and conditioning as

$$f_{\mathbf{x}^*} | \boldsymbol{x}^*, \boldsymbol{X}, \boldsymbol{y} \sim \mathcal{N}(K_{*N}(K_{NN} + \sigma_{\text{noise}}^2 I_N)^{-1}\mathbf{y}, K_{**} - K_{*N}(K_{NN} + \sigma_{\text{noise}}^2 I_N)^{-1}K_{N*}).$$

In preview of our experiments in Section 5 and the following sections, we choose a specific covariance function, the *spectral mixture* (SM) covariance function (Wilson & Adams, 2013)

$$k(\mathbf{x}, \mathbf{x}') = \sigma_{\text{power}}^2 \left( \prod_{q=1}^{Q} \exp\left( -\frac{1}{2} l_q^{-2} (x_q - x_q')^2 \right) \right) \cos\left( 2\pi \sum_{q=1}^{Q} p_q^{-1} (x_q - x_q') \right), \quad (3)$$

with an amplitude $\sigma_{\text{power}}^2$ and length scales $l_q, p_q \in \mathbb{R}$, $q = 1, \ldots, Q$. As $p_q \to \infty$, this corresponds to the *squared exponential* (SE) covariance function in the limit (Gal & Turner, 2015).

## 3.2 VARIATIONAL SPARSE SPECTRUM GP

We introduce the SSGP following Gal & Turner (2015). For a stationary covariance function $k$ on $\mathbb{R}^Q \times \mathbb{R}^Q$ there exists a function $\rho : \mathbb{R}^Q \to \mathbb{R}, \boldsymbol{\tau} \mapsto \rho(\boldsymbol{\tau})$, such that $k(\mathbf{x}, \mathbf{x}') = \rho(\mathbf{x} - \mathbf{x}')$ for $\mathbf{x}, \mathbf{x}' \in \mathbb{R}^Q$. Bochner's theorem states that any stationary covariance function $k$ can be represented as the Fourier transform of a positive finite measure $\boldsymbol{\mu}$ (Stein, 2012). Then $\rho(\boldsymbol{\tau})$, using $\boldsymbol{\tau} = \mathbf{x} - \mathbf{x}'$, can be expressed via *Monte Carlo Approximation* (MCA) following Gal & Turner (2015), Section 2, as

$$\rho(\boldsymbol{\tau}) = \int_{\mathbb{R}^Q} e^{2\pi i \mathbf{z}^T \boldsymbol{\tau}} d\boldsymbol{\mu}(\mathbf{z}) \quad (4)$$

$$\approx \frac{\sigma_{\text{power}}^2}{M} \sum_{m=1}^{M} 2 \cos(2\pi \mathbf{z}_m^T (\mathbf{x} - \mathbf{u}_m) + b_m) \cos(2\pi \mathbf{z}_m^T (\mathbf{x}' - \mathbf{u}_m) + b_m) \overset{\text{def}}{=} \tilde{k}(\mathbf{x}, \mathbf{x}').$$

We refer to $\mathbf{z}_m$ as the spectral points, $b_m$ as the spectral phases and $\mathbf{u}_m$ as the pseudo-input/inducing points for $m = 1, \ldots, M$. Choosing the probability density like in Gal & Turner (2015), Proposition 2, we approximate the SM covariance function with a scaling matrix $\mathfrak{L} \overset{\text{def}}{=} \text{diag}([2\pi l_q]_{q=1}^Q) \in \mathbb{R}^{Q \times Q}$, a scaling vector $\mathbf{p} \overset{\text{def}}{=} [p_1^{-1}, \ldots, p_Q^{-1}]^T \in \mathbb{R}^Q$ and $\mathbf{Z} \overset{\text{def}}{=} [\mathbf{z}_1, \ldots, \mathbf{z}_M]^T \in \mathbb{R}^{Q \times M}$ via

$$\phi(\mathbf{x}, \mathbf{Z}) \overset{\text{def}}{=} \sqrt{2\sigma_{\text{power}}^2 M^{-1}} \left[ \cos(2\pi (\mathfrak{L}^{-1} \mathbf{z}_1 + \mathbf{p})^T (\mathbf{x} - \mathbf{u}_1) + b_1), \ldots, \right. \quad (5)$$

$$\left. \cos(2\pi (\mathfrak{L}^{-1} \mathbf{z}_M + \mathbf{p})^T (\mathbf{x} - \mathbf{u}_M) + b_M) \right]^T \in \mathbb{R}^M.$$

Here we sample $b \sim \text{Unif}[0, 2\pi]$, $\mathbf{z} \sim \mathcal{N}(\mathbf{0}, I_Q)$ and we set $\tilde{K}_{NN}^{(\text{SM})} \overset{\text{def}}{=} \Phi \Phi^T$ with $\Phi \overset{\text{def}}{=} [\phi(\mathbf{x}_1, \mathbf{Z}), \ldots, \phi(\mathbf{x}_N, \mathbf{Z})]^T \in \mathbb{R}^{N \times M}$.

In Gal & Turner (2015) the SSGP was improved to VSSGP by variationally integrating out the spectral points and instead of optimizing the spectral points, additionally optimizing the variational parameters. We follow the scheme of Gal & Turner (2015), Section 4, for the 1-dimensional output case $\mathbf{y} \in \mathbb{R}^{N \times 1}$. By replacing the covariance function with the sparse covariance function $\tilde{k}^{(\text{SM})}$ and setting the priors to

$$p_{\mathbf{Z}} \overset{\text{def}}{=} \prod_{m=1}^{M} p_{\mathbf{z}_m}, \text{ where } \mathbf{z}_m \sim \mathcal{N}(\mathbf{0}, I_Q), \quad p_{\mathbf{a}}, \text{ where } \mathbf{a} \sim \mathcal{N}(\mathbf{0}, I_M), \quad (6)$$

for $m = 1, \ldots, M$, where we have $\mathbf{y} | \mathbf{a}, \mathbf{Z}, \mathbf{U}, \mathbf{X} \sim \mathcal{N}(\Phi \mathbf{a}, \sigma_{\text{noise}}^2 I_N)$ with $\mathbf{U} \overset{\text{def}}{=} [\mathbf{u}_1, \ldots, \mathbf{u}_M]^T \in \mathbb{R}^{Q \times M}$ (we do not define priors on $\mathbf{p}, \mathfrak{L}^{-1}, \mathbf{U}, \mathbf{b} \overset{\text{def}}{=} [b_1, \ldots, b_M]^T \in \mathbb{R}^M$), we can expand the *marginal likelihood* (ML) to

$$p(\mathbf{y}|\mathbf{X}) = \int p(\mathbf{y}|\mathbf{a}, \mathbf{Z}, \mathbf{U}, \mathbf{X}) p(\mathbf{a}) p(\mathbf{Z}) d\mathbf{a} d\mathbf{Z}, \quad (7)$$

highlighting $\mathbf{U}$ just in the integral, to be notationally conform to Gal & Turner (2015), Section 3. Now, to improve the SSGP to VSSGP, variational distributions are introduced in terms of

$$q_{\mathbf{Z}} \overset{\text{def}}{=} \prod_{m=1}^{M} q_{\mathbf{z}_m}, \text{ where } \mathbf{z}_m \sim \mathcal{N}(\boldsymbol{\alpha}_m, \boldsymbol{\beta}_m), \quad q_{\mathbf{a}}, \text{ where } \mathbf{a} \sim \mathcal{N}(\mathbf{m}, \mathbf{s}),$$

with $\boldsymbol{\beta}_m \in \mathbb{R}^{Q \times Q}$ diagonal, for $m = 1, \ldots, M$, and $\mathbf{s} \in \mathbb{R}^{M \times M}$ diagonal. From here on we use variational mean-field approximation to derive the approximate models with different lower bounds to the log ML introduced by Gal & Turner (2015):

$$\log(p(\mathbf{y}|\mathbf{X})) \geq \mathbf{E}_{q_{\mathbf{a}} q_{\mathbf{Z}}}[\log(p(\mathbf{y}|\mathbf{a}, \mathbf{Z}, \mathbf{U}, \mathbf{X}))] - \mathbf{KL}(q_{\mathbf{a}}||p_{\mathbf{a}}) - \mathbf{KL}(q_{\mathbf{Z}}||p_{\mathbf{Z}}). \quad (8)$$

As usual, $\mathbf{KL}$ defines the *Kullback-Leibler* (KL) divergence. By proving the existence of an optimal distribution $q_{\mathbf{a}}^{\text{opt}}$ for $\mathbf{a}$ Gal & Turner (2015), Proposition 3, in the sense of a functional local optimum of the right hand side of (8), where $\mathbf{a} \sim \mathcal{N}(A^{-1} \Psi_1^T \mathbf{y}, \sigma_{\text{noise}}^2 A^{-1})$, with $A = \Psi_2 + \sigma_{\text{noise}}^2 I_M$, $\Psi_1 = \mathbf{E}_{q_{\mathbf{Z}}}[\Phi] \in \mathbb{R}^{N \times M}$, $\Psi_2 = \mathbf{E}_{q_{\mathbf{Z}}}[\Phi^T \Phi] \in \mathbb{R}^{M \times M}$, we can derive the optimal bound case.

### 3.3 (V)SSGP WITH REGULARIZATION PROPERTIES VIA INDUCING POINTS (IP)

As a first contribution of this paper we combine two approximation schemes to two new methods (V)SSGP-IP. We want to point out that the (V)SSGP does not have the same regularization properties as the GP of Titsias & Lawrence (2010), when optimizing the parameters $\mathbf{U}$, because the prior in (6) of the weights $\mathbf{a}$ is defined generically via Bishop (2006), Equations (2.113) - (2.115). These parameters $\mathbf{U}$, following Gal & Turner (2015), are similar to the sparse pseudo-input approximation Titsias (2009), but in the lower bound in (8) they are simply used without being linked to the weights $\mathbf{a}$.
We now define them as

$$p_{\boldsymbol{a}|\boldsymbol{U}}, \text{ where } \boldsymbol{a}|\boldsymbol{U} \sim \mathcal{N}(\mathbf{0}, K_{MM}), \tag{9}$$

$$p_{\boldsymbol{y}|\boldsymbol{a},\boldsymbol{Z},\boldsymbol{U},\boldsymbol{X}}, \text{ where } \boldsymbol{y}|\boldsymbol{a},\boldsymbol{Z},\boldsymbol{U},\boldsymbol{X} \sim \mathcal{N}(\Phi K_{MM}^{-1}\mathbf{a}, K_{NN} - K_{NM}K_{MM}^{-1}K_{MN} + \sigma_{\text{noise}}^2 I_N), \tag{10}$$

where $K_{NN}$, $K_{NM}$, $K_{MM}$ are defined through the given covariance function in Equation (3).
We can show that for these definitions the integral in Equation (7) can be marginalized straightforward for the weights $\mathbf{a}$. We then obtain that our data-samples $\mathbf{y}$ are coming from a Gaussian distribution with mean $\mathbf{0}$ and the true covariance matrix $K_{NN}$, plus the discrepancy of the two well-known sparse covariance matrices for the Sparse Spectrum (not exactly, as we have $\Phi K_{MM}^{-1}\Phi^T$) and the Nyström case, plus the noise assumption. This expression can not be calculated efficiently, but shows that we obtain a GP approximation, which can be seen as a trade-off between these two sparse approximations.
Following Titsias & Lawrence (2010), Section 3.1, the optimal variational distribution for $\boldsymbol{a}$ collapses by reversing Jensen's inequality and is similar to the one obtained in Titsias & Lawrence (2010). The resulting bounds (SSGP-IP without $\mathbf{KL}(q_{\boldsymbol{Z}}\|p_{\boldsymbol{Z}})$ and $\Psi_1 = \Phi$, $\Psi_2 = \Phi^T\Phi$) can be calculated in the same way as Titsias & Lawrence (2010) until Equation (14) in closed form:

$$\log(p(\mathbf{y}|\mathbf{X})) \geq -\frac{(N-M)}{2}\log(\sigma_{\text{noise}}^2) - \frac{N}{2}\log(2\pi) - \frac{\mathbf{y}^T\mathbf{y}}{2\sigma_{\text{noise}}^2} + \frac{\log(|K_{MM}|\,|A^{-1}|)}{2} \tag{11}$$

$$+ \frac{\mathbf{y}^T\Psi_1 A^{-1}\Psi_1^T\mathbf{y}}{2\sigma_{\text{noise}}^2} - \frac{\text{tr}(K_{NN})}{2\sigma_{\text{noise}}^2} + \frac{\text{tr}(K_{MM}^{-1}K_{MN}K_{NM})}{2\sigma_{\text{noise}}^2} - \mathbf{KL}(q_{\boldsymbol{Z}}\|p_{\boldsymbol{Z}}).$$

Consequently, the resulting bound in (11) has an extra regularization property compared to the right hand side of (8), which is reflected in the different form of $A = \Psi_2 + \sigma_{\text{noise}}^2 K_{MM}$, which involves $K_{MM}$, the chosen covariance matrix filled in with the pseudo-input points $\mathbf{U}$, and three extra terms

$$\frac{\log(|K_{MM}|)}{2}, \quad \frac{\text{tr}(K_{NN})}{2\sigma_{\text{noise}}^2}, \quad \frac{\text{tr}(K_{MM}^{-1}K_{MN}K_{NM})}{2\sigma_{\text{noise}}^2}.$$

### 3.4 VARIATIONAL APPROXIMATION OF THE INPUT-SPACE FOR (V)SSGP(-IP)

As a second contribution of this paper we marginalize also the input-space. This is not straightforward, as it is not clear whether for the (V)SS covariance function in Equation (4) and (5) these expressions even exist. To prevent misunderstanding, we will write from now on $\mathbf{H} = [\mathbf{h}_1, \ldots, \mathbf{h}_N]^T \in \mathbb{R}^{N \times Q}$ instead of $\mathbf{X}$, to highlight that $\mathbf{H}$ is now a set of latent variables (later on the hidden states of the DGP). Therefore, we introduce priors and variational distributions

$$p_{\boldsymbol{H}} \overset{\text{def}}{=} \prod_{i=1}^{N} p_{\boldsymbol{h}_i}, \text{ where } \boldsymbol{h}_i \sim \mathcal{N}(\mathbf{0}, I_Q), \qquad q_{\boldsymbol{H}} \overset{\text{def}}{=} \prod_{i=1}^{N} q_{\boldsymbol{h}_i}, \text{ where } \boldsymbol{h}_i \sim \mathcal{N}(\boldsymbol{\mu}_i, \boldsymbol{\lambda}_i),$$

$\boldsymbol{\lambda}_i \in \mathbb{R}^{Q \times Q}$ diagonal, for $i = 1, \ldots, N$.
As a consequence, for VSSGP-IP we overall derive statistics $\Psi_0 = \text{tr}\left(\mathbf{E}_{q_H}[K_{NN}]\right) = N\sigma_{\text{power}}^2$, $\Psi_1 = \mathbf{E}_{q_Z q_H}[\Phi] \in \mathbb{R}^{N \times M}$, $\Psi_2 = \mathbf{E}_{q_Z q_H}\left[\Phi^T\Phi\right] \in \mathbb{R}^{M \times M}$ and $\Psi_{reg} = \mathbf{E}_{q_H}[K_{MN}K_{NM}]$ as defined in the Appendix 6.3. These statistics are essentially the given matrices $\Phi$, $\Phi^T\Phi$, $K_{MN}K_{NM}$ from the beginning, but every input $\mathbf{h}_i$ and every spectral point $\mathbf{z}_m$ is now replaced by a mean $\boldsymbol{\mu}_i$, $\boldsymbol{\alpha}_m$ and a variance $\boldsymbol{\lambda}_i$, $\boldsymbol{\beta}_m$ resulting in matrices of the same size. The property of positive definiteness is preserved. The SSGP-IP model derives by being not variational over the spectral points.
This extra step allows to propagate uncertainty between the hidden layers of a DGP, as we gain an

extra variance parameter for the inputs. For the IP case we get the lower bound:

$$\log(p(\mathbf{y}|\mathbf{H})) \geq -\frac{(N-M)}{2}\log(\sigma_{\text{noise}}^2) - \frac{N}{2}\log(2\pi) - \frac{\mathbf{y}^T\mathbf{y}}{2\sigma_{\text{noise}}^2} + \frac{\log(|K_{MM}||A^{-1}|)}{2} \qquad (12)$$

$$+ \frac{\mathbf{y}^T\Psi_1 A^{-1}\Psi_1^T\mathbf{y}}{2\sigma_{\text{noise}}^2} - \frac{\Psi_0}{2\sigma_{\text{noise}}^2} + \frac{\text{tr}(K_{MM}^{-1}\Psi_{reg})}{2\sigma_{\text{noise}}^2} - \mathbf{KL}(q_{\boldsymbol{Z}}||p_{\boldsymbol{Z}}) - \mathbf{KL}(q_{\boldsymbol{H}}||p_{\boldsymbol{H}}).$$

For the extension of the SSGP and VSSGP in the optimal bound case in (8) (we only focus on this case in the following), we again have $A = \Psi_2 + \sigma_{\text{noise}}^2 I_M$ and eliminate $\frac{\log(|K_{MM}|)}{2}$, $\frac{\Psi_0}{2\sigma_{\text{noise}}^2}$, $\frac{\text{tr}(K_{MM}^{-1}\Psi_{reg})}{2\sigma_{\text{noise}}^2}$ in the lower bound (12).

## 4 DRGP WITH VARIATIONAL SPARSE SPECTRUM APPROXIMATION

In this section we want to combine our newly derived (V)SS-(IP) approximations in the Sections 3.3, 3.4, overall resulting in four GP cases: SSGP, VSSGP, SSGP-IP, VSSGP-IP, with the framework introduced in Mattos et al. (2016), to derive our DRGP models: DRGP-SS, DRGP-VSS, DRGP-SS-IP, DRGP-VSS-IP.

### 4.1 DRGP-(V)SS(-IP) MODEL DEFINITION

Choosing the same recurrent structure as in Mattos et al. (2016), where now $i$ represents the time, $\mathbf{y}_{H_\mathbf{x}+1:} = [y_{H_\mathbf{x}+1}, \ldots, y_N]^T \in \mathbb{R}^{\hat{N}}$, we have

$$\zeta: \quad \mathbf{h}_i^{(l)} = \mathbf{f}^{(l)}(\hat{\mathbf{h}}_i^{(l)}) + \epsilon_i^{h^{(l)}}, \qquad \text{with prior} \quad f_{\hat{\boldsymbol{H}}^{(l)}}^{(l)}|\hat{\boldsymbol{H}}^{(l)} \sim \mathcal{N}(\mathbf{0}, K_{\hat{N}\hat{N}}^{(l)}), \qquad l = 1, \ldots, L$$

$$\psi: \quad \mathbf{y}_i = \mathbf{f}^{(l)}(\hat{\mathbf{h}}_i^{(l)}) + \epsilon_i^{y}, \qquad \text{with prior} \quad f_{\hat{\boldsymbol{H}}^{(l)}}^{(l)}|\hat{\boldsymbol{H}}^{(l)} \sim \mathcal{N}(\mathbf{0}, K_{\hat{N}\hat{N}}^{(l)}), \qquad l = L+1,$$

with $\zeta$, $\psi$ in Equation (1) and (2), $\epsilon_i^{h^{(l)}} \sim \mathcal{N}(0, (\sigma_{\text{noise}}^{(l)})^2)$, $\epsilon_i^{y} \sim \mathcal{N}(0, (\sigma_{\text{noise}}^{(L+1)})^2)$ and $\hat{N} = N - H_\mathbf{x}$, for $i = H_\mathbf{x}+1, \ldots, N$. The matrix $K_{\hat{N}\hat{N}}^{(l)}$ represents a covariance matrix coming from our chosen $\tilde{k}$ in Equation (4) and (5). A set of input-data $\hat{\mathbf{H}}^{(l)} = [\hat{h}_{H_\mathbf{x}+1}^{(l)}, \ldots, \hat{h}_N^{(l)}]^T$ is specified as

$$\hat{\mathbf{h}}_i^{(l)} \overset{\text{def}}{=} \begin{cases} \left[\overline{\mathbf{h}}_{i-1}^{(1)}, \overline{\mathbf{x}}_{i-1}\right]^T \overset{\text{def}}{=} \left[\left[\mathbf{h}_{i-1}^{(1)}, \ldots, \mathbf{h}_{i-H_h}^{(1)}\right], [\mathbf{x}_{i-1}, \ldots, \mathbf{x}_{i-H_\mathbf{x}}]\right]^T, & l = 1 \quad (13) \\[2mm] \left[\overline{\mathbf{h}}_{i-1}^{(l)}, \overline{\mathbf{h}}_i^{(l-1)}\right]^T \overset{\text{def}}{=} \left[\left[\mathbf{h}_{i-1}^{(l)}, \ldots, \mathbf{h}_{i-H_h}^{(l)}\right], \left[\mathbf{h}_i^{(l-1)}, \ldots, \mathbf{h}_{i-H_h+1}^{(l-1)}\right]\right]^T, & l = 2, \ldots, L \\[2mm] \overline{\mathbf{h}}_i^{(L) \text{def}} \left[\mathbf{h}_i^{(L)}, \ldots, \mathbf{h}_{i-H_h+1}^{(L)}\right]^T, & l = L+1, \end{cases}$$

where $\hat{\mathbf{h}}_i^{(1)} \in \mathbb{R}^{H_h + H_\mathbf{x}Q}$, $\hat{\mathbf{h}}_i^{(l)} \in \mathbb{R}^{2H_h}$ for $l = 2, \ldots, L$, $\hat{\mathbf{h}}_i^{(L+1)} \in \mathbb{R}^{H_h}$, for $i = H_\mathbf{x}+1, \ldots, N$. For simplification we set $H \overset{\text{def}}{=} H_\mathbf{x} = H_h$ in our experiments.

Now we use the new approximations in the Sections 3.3, 3.4, to derive first, for the setting (V)SSGP, the new joint probability density

$$p_{\boldsymbol{y}_{H_\mathbf{x}+1:}[\boldsymbol{a}^{(l)},\boldsymbol{Z}^{(l)},\boldsymbol{h}^{(l)},\boldsymbol{U}^{(l)}]_{l=1}^{L+1}|\boldsymbol{X}} = \prod_{l=1}^{L+1} p_{\boldsymbol{h}_{H_\mathbf{x}+1:}^{(l)}|\boldsymbol{a}^{(l)},\boldsymbol{Z}^{(l)},\hat{\boldsymbol{H}}^{(l)},\boldsymbol{U}^{(l)}} p_{\boldsymbol{a}^{(l)}} p_{\boldsymbol{Z}^{(l)}} p_{\tilde{\boldsymbol{h}}^{(l)}},$$

with $p_{\tilde{h}^{(L+1)}} = 1$, $\boldsymbol{h}_{H_\mathbf{x}+1:}^{(L+1)} \overset{\text{def}}{=} \boldsymbol{y}_{H_\mathbf{x}+1:}$, $\boldsymbol{h}^{(L+1)} = \{\}$ and $\tilde{\mathbf{h}}^{(l)} = [\mathbf{h}_{1+H_\mathbf{x}-H_h}^{(l)}, \ldots, \mathbf{h}_{H_h}^{(l)}]^T \in \mathbb{R}^{2H_h - H_\mathbf{x}}$. Here the priors are

$$p_{\boldsymbol{a}^{(l)}}, \text{ where } \boldsymbol{a}^{(l)} \sim \mathcal{N}(\mathbf{0}, I_M), \quad p_{\boldsymbol{Z}^{(l)}} \overset{\text{def}}{=} \prod_{m=1}^M p_{\boldsymbol{z}_m^{(l)}}, \text{ where } \boldsymbol{z}_m^{(l)} \sim \mathcal{N}(\mathbf{0}, I_Q),$$

$$p_{\tilde{\boldsymbol{h}}^{(l)}}, \text{ where } \tilde{\boldsymbol{h}}^{(l)} \sim \mathcal{N}(\mathbf{0}, I_{2H_h - H_\mathbf{x}}),$$

the product of them is defined as $P_{\text{REVARB}}$, and the variational distributions are

$$q_{\boldsymbol{a}^{(l)}}, \text{ where } \boldsymbol{a}^{(l)} \sim \mathcal{N}(\mathbf{m}^{(l)}, \mathbf{s}^{(l)}), \quad q_{\boldsymbol{Z}^{(l)}} \overset{\text{def}}{=} \prod_{m=1}^M q_{\boldsymbol{z}_m^{(l)}}, \text{ where } \boldsymbol{z}_m^{(l)} \sim \mathcal{N}(\boldsymbol{\alpha}_m^{(l)}, \boldsymbol{\beta}_m^{(l)}),$$

$$q_{\boldsymbol{h}^{(l)}} \overset{\text{def}}{=} \prod_{i=1+H_h-H_\mathbf{x}}^N q_{h_i^{(l)}}, \text{ where } h_i^{(l)} \sim \mathcal{N}(\mu_i^{(l)}, \lambda_i^{(l)}),$$

the product of them is defined as $Q_{\text{REVARB}}$, and where $\boldsymbol{\beta}_m^{(l)} \in \mathbb{R}^{Q \times Q}$ is diagonal, for $i = 1 + H_\mathbf{x} - H_\mathrm{h}, \ldots, N, m = 1 \ldots, M, l = 1, \ldots, L+1$.

For the setting (V)SSGP-IP we choose no variational distribution for $\boldsymbol{a}^{(l)}$, but, similar to the assumptions in (9), (10) for $l = 1, \ldots, L+1$, the prior assumptions

$$p_{\boldsymbol{a}^{(l)}|\boldsymbol{U}^{(l)}}, \text{ where } \boldsymbol{a}^{(l)}|\boldsymbol{U}^{(l)} \sim \mathcal{N}(\mathbf{0}, K_{MM}^{(l)}),$$

$$p_{\boldsymbol{h}_{H_\mathbf{x}+1:}^{(l)}|\boldsymbol{a}^{(l)}, \boldsymbol{Z}^{(l)}, \hat{\boldsymbol{H}}^{(l)}, \boldsymbol{U}^{(l)}}, \text{ where } \boldsymbol{h}_{H_\mathbf{x}+1:}^{(l)}|\boldsymbol{a}^{(l)}, \boldsymbol{Z}^{(l)}, \hat{\boldsymbol{H}}^{(l)}, \boldsymbol{U}^{(l)} \sim$$

$$\mathcal{N}(\Phi^{(l)}(K_{MM}^{(l)})^{-1}\mathbf{a}^{(l)}, K_{NN}^{(l)} - K_{NM}^{(l)}(K_{MM}^{(l)})^{-1}K_{MN}^{(l)} + (\sigma_{\text{noise}}^{(l)})^2 I_N).$$

This defines our models for the cases DRGP-VSS, DRGP-VSS-IP. In the case, where we are not variational over the spectral-points, we derive the simplified versions DRGP-SS, DRGP-SS-IP.

## 4.2 DRGP-(V)SS(-IP) VARIATIONAL EVIDENCE LOWER BOUND (ELBO)

Using standard variational approximation techniques (Blei et al., 2017), the *Recurrent Variational Bayes* lower bound for the *(V)SS* approximations, denoted as REVARB-(V)SS, is given by

$$\log(p(\mathbf{y}_{H_\mathbf{x}+1:}|\mathbf{X})) \geq$$
$$\mathbf{E}_{Q_{\text{REVARB}}}\left[\sum\nolimits_{l=1}^{L+1} \log(p(\mathbf{h}_{H_\mathbf{x}+1:}^{(l)}|\mathbf{a}^{(l)}, \mathbf{Z}^{(l)}, \hat{\mathbf{H}}^{(l)}, \mathbf{U}^{(l)}))\right] - \mathbf{KL}(Q_{\text{REVARB}}||P_{\text{REVARB}}) \overset{\text{def}}{=} \mathcal{L}_{\text{(V)SS}}^{\text{REVARB}}, \quad (14)$$

and for the *(V)SS-IP* approximations, denoted as REVARB-(V)SS-IP, is given by

$$\log(p(\mathbf{y}_{H_\mathbf{x}+1:}|\mathbf{X})) \geq$$
$$\mathbf{E}_{P_{\boldsymbol{a}^{(l)}|\boldsymbol{U}^{(l)}}}\left[\sum\nolimits_{l=1}^{L+1} \exp(\langle \log(\mathcal{N}(\mathbf{h}_{H_\mathbf{x}+1:}^{(l)}|\Phi^{(l)}(K_{MM}^{(l)})^{-1}\mathbf{a}^{(l)}, (\sigma_{\text{noise}}^{(l)})^2 I_N))\rangle_{Q_{\text{REVARB}}})\right]$$
$$- \sum\nolimits_{l=1}^{L+1} \frac{\Psi_0^{(l)}}{2(\sigma_{\text{noise}}^{(l)})^2} - \frac{\text{tr}((K_{MM}^{(l)})^{-1}\Psi_{reg}^{(l)})}{2(\sigma_{\text{noise}}^{(l)})^2} - \mathbf{KL}(Q_{\text{REVARB}}||P_{\text{REVARB}}) \overset{\text{def}}{=} \mathcal{L}_{\text{(V)SS(-IP)}}^{\text{REVARB}}, \quad (15)$$

where $\langle \cdot, \cdot \rangle$ means, taking the expectation under the integral.

Additionally, for the simple (V)SS approximations in 14, the optimal bound $\mathcal{L}_{\text{(V)SS-opt}}^{\text{REVARB}}$ can be obtained immediately, analogously to Gal & Turner (2015), Proposition 3, and by the fact that the bound decomposes into a sum of independent terms for $\boldsymbol{a}^{(l)}$. Maximizing the lower bounds is equivalent to minimizing the KL-divergence of $Q_{\text{REVARB}}$ and the true posterior. Therefore, this is a way to optimize the approximated model parameter distribution with respect to the intractable, true model parameter posterior. Calculating $\mathcal{L}_{\text{(V)SS-opt/IP}}^{\text{REVARB}}$ requires $\mathcal{O}(NM^2 Q_{\max}(L+1))$, where $Q_{\max} = \max_{l=1\ldots,L+1} Q^{(l)}$, $Q^{(l)} \overset{\text{def}}{=} \dim(\hat{\mathbf{h}}_i^{(l)})$ and $\hat{\mathbf{h}}_i^{(l)}$ from Equation (13). DVI can reduce the complexity to $\mathcal{O}(M^3)$ if the number of cores scales suitably with the number of training-data, see Appendix 6.3.6 for a detailed description. A detailed derivation of the REVARB-(V)SS(-IP) lower bounds can be found in the Appendix 6.3.4.

## 4.3 MODEL PREDICTIONS

After model optimization based on the lower bounds in the Equation (14) and (15), model predictions for new $\hat{\mathbf{h}}_*^{(l)}$ in the REVARB-(V)SS(-IP) framework can be obtained based on the approximate, variational posterior distribution $Q_{\text{REVARB}}$. They are performed iteratively with approximate uncertainty propagation between each GP layer. We derive $q_{\boldsymbol{h}_*^{(l)}}$ from previous time-steps and it is per definition Gaussian with mean and variance derived from previous predictions for $l = 1, \ldots, L$. These kind of models propagate the uncertainty of the hidden GP layers' outputs, not of the observed output-data and are relevant for good model predictions. The detailed expressions for the mean and variance of the predictive distribution involved during the predictions can be found in the Appendix 6.3.5.

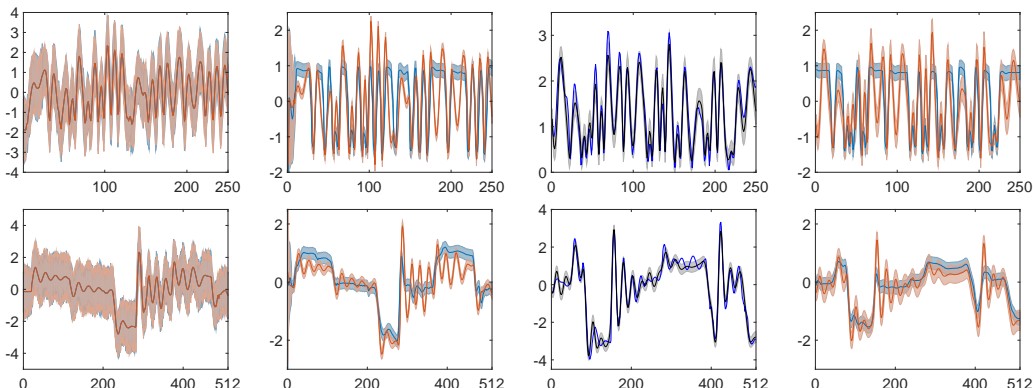

Figure 1: Simulation results visualized for the data-sets Drive (first row) and Actuator (second row) for the method DRGP-VSS. First column represents the initial hidden states, blue: first layer, and red: second layer. The second column represents the learned hidden states. The third column shows the simulation results, blue: real data, black: simulation, grey: $\pm 2$ times Standard Deviation (SD) and the fourth column the predicted hidden states.

## 5 EXPERIMENTS

In this section we want to compare our methods DRGP-SS, DRGP-VSS (optimal bound case) and DRGP-SS-IP, DRGP-VSS-IP, against other well known sparse GPs and the full GP with NARX structure, the DRGP-Nyström of Mattos et al. (2016), the GP-LSTM of Al-Shedivat et al. (2017), the LSTM of Hochreiter & Schmidhuber (1997), a simple RNN, the DGP-DS of Salimbeni & Deisenroth (2017) and the DGP-RFF of Cutajar et al. (2016), both with NARX structure for the first layer, the GP-SSM of Svensson et al. (2016) and the PR-SSM Dörr et al. (2018). The full GP is named GP-full, the FITC approximation of Snelson & Ghahramani (2006) is named GP-FITC, the DTC approximation of Williams & Seeger (2000) is named GP-DTC, the SSGP of Lázaro-Gredilla et al. (2010) is named GP-SS, the VSSGP of Gal & Turner (2015) is named GP-VSS. The setting in this system identification task is simulation. This means that, together with past exogenous inputs, no past measured output observations (but perhaps predicted output observations) are used to predict next output observations. To enable a fair comparison, all methods are given access to a predefined amount of data. Details about the methods, their configuration, as well as the benchmark data-sets can be found in the Appendix 6.1 and 6.2.

### 5.1 IMPLEMENTATION

Our methods DRGP-(V)SS(-IP) were implemented in Python, using the library Theano, and in Matlab R2016b. For the optimization/training we used Python, Theano. Theano allows us to take full advantage of the automatic differentiation to calculate the gradients. For simulation and visualization we used Matlab R2016b.

We further implemented in Matlab R2016b the methods DRGP-Nyström, GP-SS, GP-DTC, GP-FITC, GP-full and used these implementations for the experiments. For GP-VSS, GP-LSTM, LSTM, RNN, DGP-RFF and DGP-DS we used the published code[3456]. For GP-SSM the published code is only applicable for small range of state dimension and a small time horizon, so we just show the results from their paper. For PR-SSM we also show the results from their paper.

---

[3]GP-VSS code available from `https://github.com/yaringal/VSSGP`.

[4]GP-LSTM, LSTM, RNN code available from `https://github.com/alshedivat/keras-gp`.

[5]DGP-DS code available from `https://github.com/ICL-SML/Doubly-Stochastic-DGP`.

[6]DGP-RFF code available from `https://github.com/mauriziofilippone/deep_gp_random_features`.

Table 1: Summary of RMSE values for the free simulation results on test data. Best values per data-set are bold. All values are calculated on the original data, unless the data-set Power Load, where the RMSE is shown for the normalized data. Here we have full recurrence for our methods, DRGP-Nyström and GP-LSTM, LSTM, RNN and with auto-regressive part (first layer) for all other GPs. For the column non-rec we turned off the auto-regressive part in the first layer for our methods, DRGP-Nyström and GP-LSTM, LSTM, RNN and also turned off the auto-regressive part (first layer) for all other GPs.

| methods-data | Emission | non-rec | Power Load | Damper | Actuator | non-rec | Ballbeam | Dryer | Drive | non-rec |
|---|---|---|---|---|---|---|---|---|---|---|
| **DRGP-VSS** | *0.104* | *0.062* | ***0.457*** | *5.825* | *0.357* | ***0.388*** | *0.084* | *0.109* | *0.229* | *0.268* |
| **DRGP-VSS-IP** | *0.119* | *0.064* | *0.544* | *6.112* | *0.441* | *0.546* | *0.071* | *0.107* | *0.302* | *0.293* |
| **DRGP-SS** | *0.108* | *0.062* | *0.497* | *5.277* | ***0.329*** | *0.563* | *0.081* | *0.108* | ***0.226*** | ***0.253*** |
| **DRGP-SS-IP** | *0.118* | *0.065* | *0.631* | ***5.129*** | *0.534* | *0.547* | *0.076* | *0.107* | *0.297* | *0.261* |
| DRGP-Nyström | 0.109 | 0.059 | 0.493 | 6.344 | 0.368 | 0.415 | 0.082 | 0.109 | 0.249 | 0.289 |
| GP-LSTM | 0.096 | 0.091 | 0.529 | 9.083 | 0.430 | 0.730 | **0.062** | 0.108 | 0.320 | 0.530 |
| LSTM | 0.098 | 0.061 | 0.530 | 9.370 | 0.440 | 0.640 | **0.062** | 0.090 | 0.400 | 0.570 |
| RNN | 0.098 | 0.066 | 0.548 | 9.012 | 0.680 | 0.690 | 0.063 | 0.121 | 0.560 | 0.590 |
| DGP-DS | 0.106 | 0.062 | 0.543 | 6.267 | 0.590 | 0.576 | 0.066 | **0.085** | 0.422 | 0.571 |
| DGP-RFF | **0.092** | 0.069 | 0.550 | 5.415 | 0.520 | 0.750 | 0.074 | 0.093 | 0.446 | 0.732 |
| PR-SSM | N/A | N/A | N/A | N/A | 0.502 | N/A | 0.073 | 0.140 | 0.492 | N/A |
| GP-SSM | N/A | N/A | N/A | 8.170 | N/A | N/A | N/A | N/A | N/A | N/A |
| GP-VSS | 0.130 | 0.058 | 0.514 | 6.554 | 0.449 | 0.767 | 0.120 | 0.112 | 0.401 | 0.549 |
| GP-SS | 0.128 | 0.060 | 0.539 | 6.730 | 0.439 | 0.777 | 0.077 | 0.106 | 0.358 | 0.556 |
| GP-DTC | 0.137 | 0.061 | 0.566 | 7.474 | 0.458 | 0.864 | 0.122 | 0.105 | 0.408 | 0.540 |
| GP-FITC | 0.126 | **0.057** | 0.536 | 6.754 | 0.433 | 0.860 | 0.084 | 0.108 | 0.403 | 0.539 |
| GP-full | 0.122 | 0.066 | 0.696 | 9.890 | 0.449 | 1.037 | 0.128 | 0.106 | 0.444 | 0.542 |

## 5.2 BENCHMARK DATA-SETS MODEL LEARNING AND COMPARISON

In Figure 1 we show a comparison of the latent model states before and after training, the simulation results, as well as the simulated latent states for two data-sets, Drive and Actuator, for the model DRGP-VSS. We initialize the states with the output training-data for all layers with minor noise (first column) and after training we obtain a trained state (second column). Unlike Mattos et al. (2016) Figure 2, (l), we get good variance predictions for all data-sets. We used our own implementation for Mattos et al. (2016), which gave the same good results as for our methods. Therefore, we think it is an implementation issue. The test RMSE results for all methods are summarized in Table 1. The results show, that on most data-sets DRGP-(V)SS(-IP) improve slightly in comparison to other methods. In order to evaluate the reproducing quality of our results, we provide a robustness test for our methods and DRGP-Nyström on the data-sets Drive and Damper in Figure 2, 3. We run the optimization for different time horizons. For every method we visualized a boxplot with whiskers from minimum to maximum with 10 independent runs. For our models we obtain good results compared with DRGP-Nyström on these data-sets, in particular for the setting of time horizons of Table 1 with $H = 10$. In Figure 4 the RMSE results for different layers $L$ on the data-sets Drive, Actuator and Damper are shown. We can observe that different layers $L$ are favored.

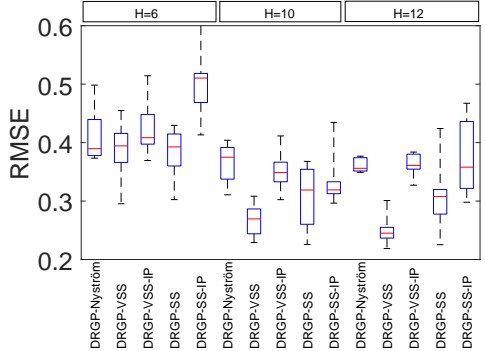
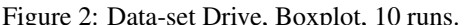

Figure 2: Data-set Drive, Boxplot, 10 runs.

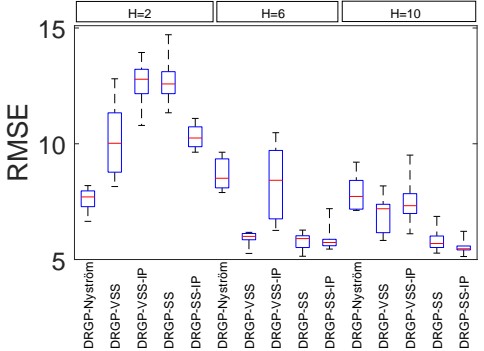

Figure 3: Data-set Damper, Boxplot, 10 runs.

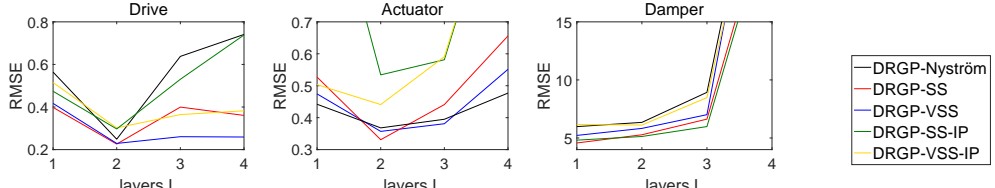

Figure 4: RMSE values for different number of layers L for the data-sets Drive, Actuator, Damper.

## 5.3 CONCLUSION

In this paper we introduced four new DRGPs based on the SS approximation introduced by Lázaro-Gredilla et al. (2010) and the improved VSS approximation by Gal & Turner (2015). We combined the (V)SS approximations with the variational inducing-point approximation from Titsias & Lawrence (2010), also integrated variationally over the input-space and established the existence of an optimal variational distribution for $a^{(l)}$ in the sense of a functional local optimum of the variational bounds REVARB-(V)SS-(IP). We could show that our methods slightly improve on the data-sets used in this paper compared to the RGP from Mattos et al. (2016) and other state-of-the-art methods, where moreover our sparse approximations are also practical for dimensionality reduction as shown in Titsias & Lawrence (2010) and can be further expanded to a deep version in this application (Damianou & Lawrence, 2013). Furthermore, Hoang et al. (2017) introduced a generalized version of the (V)SS approximation, which should be adaptable for our case.

ACKNOWLEDGMENTS

We would like to acknowledge support for this project from the ETAS GmbH.

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

## 6 APPENDIX

This additional material provides details about the derivations and configurations of the proposed DRGP-(V)SS(-IP) in the Sections 3-4. and elaborates on the methods and the employed data-sets in the experiments in Section 5.

### 6.1 DATA-SET DESCRIPTION

Table 2: Summary of the data-sets for the system identification tasks

| parameters \ data-sets | $N$ | $N_{train}$ | $N_{test}$ | input dimension | output dimension |
|---|---|---|---|---|---|
| Drive | 500 | 250 | 250 | 1 | 1 |
| Dryer | 1000 | 500 | 500 | 1 | 1 |
| Ballbeam | 1000 | 500 | 500 | 1 | 1 |
| Actuator | 1024 | 512 | 512 | 2 | 1 |
| Damper | 3499 | 2000 | 1499 | 1 | 1 |
| Power Load | 9518 | 7139 | 2379 | 11 | 1 |
| Emission | 12500 | 10000 | 2500 | 6 | 1 |

In this section we introduce the data-sets we used in our experiments. We chose a large number of data-sets in training size going from 250 to 12500 data-points in order to show the performance for a wide range. We will begin with the smallest, the *Drive* data-set, which was first introduced by Wigren (2010). It is based on a system which has two electric motors that drive a pulley using a flexible belt. The input is the sum of voltages applied to the motors and the output is the speed of the belt. The data-set *Dryer*[7] describes a system where air is fanned through a tube and heated at an inlet. The input is the voltage over the heating device (a mesh of resistor wires). The output is the air temperature measured by a thermocouple. The third data-set *Ballbeam*[7][8] describes a system where the input is the angle of a beam and the output the position of a ball. *Actuator* is the name of the fourth data-set, which was described by Sjöberg et al. (1995) and which stems from an hydraulic actuator that controls a robot arm, where the input is the size of the actuator's valve opening and the output is its oil pressure. The *Damper* data-set, introduced by Wang et al. (2009), poses the problem of modeling the input–output behavior of a magneto-rheological fluid damper and is also used as a case study in the System Identification Toolbox of Mathworks Matlab. The data-set *Power Load*[9], used in Al-Shedivat et al. (2017), consists of data where the power load should be predicted from the historical temperature data. This data-set was used for 1-step ahead prediction, where past measured output observations are used to predict current or next output observations, but we will use it here for free simulation. We down-sampled by starting with the first sample and choosing every 4th data-point, because the original data-set with a size of 38064 samples and a chosen time-horizon of 48 is too large for our implementation, which is not parallelized so far. The newly provided data-set *Emission*[10] contains an emission-level of nitrogen oxide from a driving car as output and as inputs the indicated torque, boost pressure, EGR (exhaust gas recirculation) rate, injection, rail pressure and speed. The numerical characteristics of all data-sets are summarized in Table 2. The separation of the data-sets Drive, Dryer, Ballbeam, Actuator, Damper, Power Load in training- and test-data was given by the papers we use for comparison. The separation of the Emission data-set was chosen by ourself.

---

[7]Received from `http://homes.esat.kuleuven.be/~tokka/daisydata.html`.
[8]Description can be found under `http://forums.ni.com/t5/NI-myRIO/myBall-Beam-Classic-Control-Experiment/ta-p/3498079`.
[9]Originally received from Global Energy Forecasting Kaggle competitions organized in 2012.
[10]Available from `http://github.com/RomanFoell/DRGP-VSS`.

## 6.2 NONLINEAR SYSTEM IDENTIFICATION

Table 3: Summary of the methods for the system identification tasks

| method | references |
|---|---|
| DRGP-VSS | introduced in this paper |
| DRGP-VSS-IP | introduced in this paper |
| DRGP-SS | introduced in this paper |
| DRGP-SS-IP | introduced in this paper |
| DRGP-Nyström | (Mattos et al., 2016) |
| GP-LSTM | (Al-Shedivat et al., 2017) |
| LSTM | (Hochreiter & Schmidhuber, 1997) |
| RNN | see e.g. (Al-Shedivat et al., 2017) |
| DGP-DS | (Salimbeni & Deisenroth, 2017) |
| DGP-RFF | Cutajar et al. (2016) |
| PR-SSM | (Dörr et al., 2018) |
| GP-SSM | (Svensson et al., 2016) |
| GP-VSS | (Gal & Turner, 2015) |
| GP-SS | (Lázaro-Gredilla et al., 2010) |
| GP-DTC | (Williams & Seeger, 2000) |
| GP-FITC | (Snelson & Ghahramani, 2006) |
| GP-full | (Rasmussen, 2006) |

The methods for the system identification tasks and their references are summarized in Table 2. For the data-sets Drive and Actuator we chose for our methods DRGP-(V)SS(-IP) the setting $L = 2$ hidden layers, $M = 100$ spectral-points and time-horizon $H_\mathrm{h} = H_\mathbf{x} = 10$, which was also used in Mattos et al. (2016), Al-Shedivat et al. (2017) and Dörr et al. (2018) for free simulation (using $M$ pseudo-input points instead of spectral-points). For these two data-sets we filled the results from Mattos et al. (2016); Al-Shedivat et al. (2017) into Table 1. Further, for our methods DRGP-(V)SS(-IP) and DRGP-Nyström we chose on the data-sets Ballbeam and Dryer $L = 1$, $M = 100$ and $H_\mathrm{h} = H_\mathbf{x} = 10$. For the data-set Damper we chose $L = 2$, $M = 125$ and $H_\mathrm{h} = H_\mathbf{x} = 10$. For the data-set Power Load we chose $L = 1$, $M = 125$ and $H_\mathrm{h} = H_\mathbf{x} = 12$. For the data-set Emission we chose $L = 1$, $M = 125$ and $H_\mathrm{h} = H_\mathbf{x} = 10$.

The other sparse GP, the full GP, DGP-RFF and DGP-DS were trained with NARX structure $H_\mathbf{x} = H_\mathrm{y}$ (for the first layer) with the same time horizon as for our DRGPs and with the same amount of pseudo-input points or spectral points.

For LSTM, RNN we chose the same setting for the amount of hidden layers and time horizon as for our DRGPs. We used Adam optimizer with a learning rate of $0.01$. As activation function we chose $\tanh$ and NARX structure $H_\mathbf{x} = H_\mathrm{y}$ for the first layer. We tested with 8, 16, 32, 64, 128 hidden units (every hidden layer of a RNN is specified by a hidden unit parameter) for all training data-sets. For GP-LSTM we chose the same setting for the amount of hidden layers, pseudo-input points and time horizon as for our DRGPs. We used Adam optimizer with a learning rate of $0.01$. As activation function we chose $\tanh$ and NARX structure $H_\mathbf{x} = H_\mathrm{y}$ for the first layer. We tested with 8, 16, 32, 64, 128 hidden units (every hidden layer of a RNN is specified by a hidden unit parameter) for all training data-sets. For the data-sets with training size smaller or equal to 2000 we chose the version GP-LSTM in Al-Shedivat et al. (2017) and for the ones larger than 2000 the scalable version MSGP-LSTM. We did not pre-train the weights.

For DGP-RFF we tested for $L = 1, \dots, 3$ number of GPs. We used Adam optimizer with a learning rate of $0.01$. The batch size was chosen to be the training data-size and the dimensions of the GP layers to be 5. We set the flags q_Omega_fixed=1000 and learn_Omega_fixed=var_fixed, so fixing the spectral-point parameters, and iterated until 2000 just for this GP.

For DGP-DS we tested for $L = 1, \dots, 3$ number of GPs. We used natural gradients for the last GP with gamma 0.1 and Adam optimizer for the others with a learning rate of $0.01$. The batch size was chosen to be the training data-size and the dimensions of the GP layers to be 5.

All GPs which use the Nyström approximation were initialized for the pseudo-inputs points with a random subset of size $M$ from the input-data and trained with SE covariance function. For the ones which use the (V)SS approximations, which includes our methods, we trained with a spectral-point initialization sampled from $\mathcal{N}(\mathbf{0}, I_{Q^{(l)}})$, an initialization for the pseudo-input points with a random subset of size $M$ from the input-data (or setting them all to zero; not in the IP case). For our methods DRGP-(V)SS(-IP) and GP-VSS we fixed the length scales $\mathrm{p}_q^{(l)} = \infty$, for all $q, l$. So all GPs with

(V)SS approximations were also initialized as SE covariance function, see Equation 5.

For all methods we used automatic relevance determination, so each input dimension has its own length scale. For our methods DRGP-(V)SS(-IP) and DRGP-Nyström the noise parameters and the hyperparameters were initialized by $\sigma_{\text{noise}}^{(l)} = 0.1$, $\sigma_{\text{power}}^{(l)} = 1$ and the length scales by either $l_q^{(l)} = \sqrt{\max(\hat{\mathbf{H}}_q^{(l)}) - \min(\hat{\mathbf{H}}_q^{(l)})}$ or $l_q^{(l)} = \max(\hat{\mathbf{H}}_q^{(l)}) - \min(\hat{\mathbf{H}}_q^{(l)})$, for all $q, l$, where $\hat{\mathbf{H}}_q^{(l)}$ is the data-vector containing the $q$-th input-dimension values of every input-data point $\hat{\mathbf{h}}_i^{(l)}$, for all $i$. Furthermore, we initialized the latent hidden GP states with the output-observation data $\mathbf{y}$.

The other standard GPs were also initialized with $\sigma_{\text{noise}} = 0.1$, $\sigma_{\text{power}} = 1$ and the same initialization for length scales with respect to the NARX structure input data as before.

For LSTM, RNN we used the initialization for the weights provided by Keras, a Deep Learning library for Theano and TensorFlow.

For GP-LSTM we used the initialization for the weights provided by Keras and $\sigma_{\text{noise}} = 0.1$, $\sigma_{\text{power}} = 1$ and for the length scale initialization we chose $l_q = 1$ for all input-dimensions.

For DGP-RFF we used the initialization coming from the implementation with $\sigma_{\text{power}}^{(l)} = 1$ and for the length scale initialization with $l_q^{(l)} = 0.5 \log(Q^{(l)})$ for all $l, q$. To our knowledge DGP-RFF has no parameter $\sigma_{\text{noise}}^{(l)}$ for all $l$.

For DGP-DS we used the initialization $\sigma_{\text{noise}}^{(l)} = 0.1$ and $\sigma_{\text{power}}^{(l)} = 1$ and for the length scale initialization we chose $l_q^{(l)} = 1$ for all $l, q$.

For all our implementations, see Section 5.1, we used the positive transformation $x' = \log(1 + \exp(x))^2$ for the calculation of the gradients in order for the parameters to be constrained positive and with L-BFGS optimizer, either from Matlab R2016b with fmincon, or Python 2.7.12 with scipy optimize.

All methods were trained on the normalized data $x \mapsto \frac{x - \mu}{\sigma^2}$ (for every dimension independently), several times (same amount per data-set: the initializations are still not deterministic, e.g. for pseudo-inputs points and spectral points) with about 50 to 100 iterations and the best results in RMSE value on the test-data are shown in Table 1.

For our methods DRGP-(V)SS(-IP) and DRGP-Nyström we fixed $\sigma_{\text{noise}}^{(l)}$, $\sigma_{\text{power}}^{(l)}$ for all $l$ (optional the spectral points/pseudo-input points for DRGP-(V)SS; for the IP case we never excluded the pseudo-input points because we would fall back to the DRGP-(V)SS case; for DRGP-Nyström we always included the pseudo-input points) during the first iterations to independently train the latent states. For all other GPs we also tested with fixed and not fixed $\sigma_{\text{noise}} = 0.1$, except GP-LSTM and DGP-DS. For DRGP-VSS(-IP) we fixed $\boldsymbol{\beta}_m^{(l)}$ for all $m, l$ to small value around $0.001$, as well as the spectral phases $b_m$ for all $m, l$ sampling from Unif $[0, 2\pi]$ (this seems to work better in practice). The limitations for $\boldsymbol{\beta}_m^{(l)}$ also holds for GP-VSS as well.

We want to remark at this point that setting $\mathbf{u}_m = \mathbf{0}$ for all $m = 1, \ldots, M$ worked sometimes better than choosing a subset from the input-data (not in the IP case). This seems to be different to Gal & Turner (2015), who pointed out: 'These are necessary to the approximation. Without these points (or equivalently, setting these to $\mathbf{0}$), the features would decay quickly for data points far from the origin (the fixed point $\mathbf{0}$).'

For GP-SSM we show the result of the data-set Damper from their paper in Table 1.

For PR-SSM we show the results from their paper in Table 1.

We show additional results for the data-sets Drive, Actuator and Emission with missing auto-regressive part for the first layer for our methods DRGP-(V)SS(-IP) and DRGP-Nyström in Table 1, named non-rec. For the sparse GP, the full GP, GP-LSTM, DGP-RFF and DGP-DS and the data-sets Drive, Actuator and Emission we show the results with missing auto-regressive part in Table 1, just modeling the data with exogenous inputs. Here we want to examine the effect of the auto-regressive part of the first layer for the DRGP models on the RMSE. GP-SSM and PR-SSM are not listed for this setting of recurrence.

## 6.3 VARIATIONAL APPROXIMATION AND INFERENCE FOR DRGP-(V)SS(-IP)

In the Sections 6.3.1-6.3.3 we derive the statistics $\Psi_0$, $\Psi_1$, $\Psi_2$ and $\Psi_{reg}$ for the four model cases DRGP-(V)SS(-IP). In Section 6.3.4 we derive the resulting variational lower bounds. In Section 6.3.5 we show the mean and variance expressions of the predictive distributions of our DRGP models.

In the following we use the abbreviations and formulas

- B.1, (Gal & Turner, 2015, see Section 4.1),

- B.2, (Rasmussen, 2006, see A.7.),

- $\frac{1}{2}(\cos(a - b + x - y) + \cos(a + b + x + y)) = \cos(x + a)\cos(y + b)$.

- JI, for Jensen's inequality.

### 6.3.1 DRGP-VSS(-IP), THE STATISTICS $\Psi_1$, $\Psi_2$

For the versions DRGP-VSS(-IP) the statistics are

$$
\begin{aligned}
(\Psi_1)_{nm} &= \mathbf{E}_{q_{\mathbf{z}_m} q_{\mathbf{h}_n}}\left[\Phi_{nm}\right] \\
&\overset{\text{B.1}}{=} \mathbf{E}_{q_{\mathbf{h}_n}}\left[\sqrt{2\sigma_{\text{power}}^2 (M)^{-1}}e^{-\frac{1}{2}\bar{\mathbf{h}}_{nm}^T \boldsymbol{\beta}_m \bar{\mathbf{h}}_{nm}} \cos(\hat{\boldsymbol{\alpha}}_m^T(\mathbf{h}_n - \mathbf{u}_m) + \mathrm{b}_m)\right] \\
&\overset{\text{B.2}}{=} \Sigma_m^1 Z_{nm} e^{-\frac{1}{2}\hat{\boldsymbol{\alpha}}_m^T C_{nm}\hat{\boldsymbol{\alpha}}_m} \cos(\hat{\boldsymbol{\alpha}}_m^T(\mathbf{c}_{nm} - \mathbf{u}_m) + \mathrm{b}_m),
\end{aligned}
$$

for $m, = 1, \ldots, M$, $n = 1, \ldots, N$ with

$$
\begin{aligned}
\bar{\mathbf{h}}_{nm} &= 2\pi \mathfrak{L}^{-1}(\mathbf{h}_n - \mathbf{u}_m), \\
\hat{\boldsymbol{\alpha}}_m &= 2\pi(\mathfrak{L}^{-1}\boldsymbol{\alpha}_m + \mathbf{p}), \\
\mathbf{c}_{nm} &= \mathrm{C}_{nm}(\boldsymbol{\beta}_m(2\pi)^2 \mathfrak{L}^{-2}\mathbf{u}_m + \boldsymbol{\lambda}_n^{-1}\boldsymbol{\mu}_n), \\
\mathrm{C}_{nm} &= (\boldsymbol{\beta}_m(2\pi)^2 \mathfrak{L}^{-2} + \boldsymbol{\lambda}_n^{-1})^{-1}, \\
\mathbf{v}_{nm} &= \mathbf{u}_m - \boldsymbol{\mu}_n, \\
\mathrm{V}_{nm} &= (2\pi)^{-2}\mathfrak{L}^2 \boldsymbol{\beta}_m^{-1} + \boldsymbol{\lambda}_n, \\
\mathrm{Z}_{nm} &= \frac{1}{\sqrt{|\mathrm{V}_{nm}|}}e^{-\frac{1}{2}\mathbf{v}_{nm}^T \mathrm{V}_{nm}^{-1}\mathbf{v}_{nm}}, \\
\Sigma_m^1 &= \sqrt{2\sigma_{\text{power}}^2 \prod_{q=1}^Q \left(\frac{\mathrm{l}_q^2}{\boldsymbol{\beta}_{m_q}}\right)(M)^{-1}},
\end{aligned}
$$

and

$\Psi_2 = \sum_{n=1}^N (\Psi_2)^n$, where

$$
\begin{aligned}
(\Psi_2)_{mm'}^n &= \mathbf{E}_{q_{\mathbf{z}_m} q_{\mathbf{z}_{m'}} q_{\mathbf{h}_n}}\left[\Phi_{nm}^T \Phi_{nm'}\right] \\
&\overset{\text{B.1}}{=} \mathbf{E}_{q_{\mathbf{h}_n}}\Big[2\sigma_{\text{power}}^2 (M)^{-1}e^{-\frac{1}{2}(\bar{\mathbf{h}}_{nm}^T \boldsymbol{\beta}_m \bar{\mathbf{h}}_{nm} + \bar{\mathbf{h}}_{nm'}^T \boldsymbol{\beta}_{m'} \bar{\mathbf{h}}_{nm'})} \\
&\quad \cos(\hat{\boldsymbol{\alpha}}_m^T(\mathbf{h}_n - \mathbf{u}_m) + \mathrm{b}_m)\cos(\hat{\boldsymbol{\alpha}}_{m'}^T(\mathbf{h}_n - \mathbf{u}_{m'}) + \mathrm{b}_{m'})\Big] \\
&\overset{\text{B.2}}{=} \Sigma_{mm'}^2 \mathrm{Z}_{mm'}^n \Big(e^{-\frac{1}{2}\bar{\boldsymbol{\alpha}}_{mm'}^T \mathrm{D}_{mm'}^n \bar{\boldsymbol{\alpha}}_{mm'}} \cos(\bar{\boldsymbol{\alpha}}_{mm'}^T \mathbf{d}_{mm'}^n - \bar{\tau}_{mm'} + \bar{\mathrm{b}}_{mm'}) \\
&\quad + e^{-\frac{1}{2}\overset{+}{\boldsymbol{\alpha}}_{mm'}^T \mathrm{D}_{mm'}^n \overset{+}{\boldsymbol{\alpha}}_{mm'}} \cos(\overset{+}{\boldsymbol{\alpha}}_{mm'}^T \mathbf{d}_{mm'}^n - \overset{+}{\tau}_{mm'} + \overset{+}{\mathrm{b}}_{mm'})\Big).
\end{aligned}
$$

for $m, m' = 1, \ldots, M$, $m \neq m'$, with

$$\overline{\mathrm{b}}_{mm'} = \mathrm{b}_m - \mathrm{b}_{m'},$$

$$\overset{+}{\mathrm{b}}_{mm'} = \mathrm{b}_m + \mathrm{b}_{m'},$$

$$\bar{\tau}_{mm'} = \hat{\boldsymbol{\alpha}}_m^T \mathbf{u}_m - \hat{\boldsymbol{\alpha}}_{m'}^T \mathbf{u}_{m'},$$

$$\overset{+}{\tau}_{mm'} = \hat{\boldsymbol{\alpha}}_m^T \mathbf{u}_m + \hat{\boldsymbol{\alpha}}_{m'}^T \mathbf{u}_{m'},$$

$$\bar{\boldsymbol{\alpha}}_{mm'} = \hat{\boldsymbol{\alpha}}_m - \hat{\boldsymbol{\alpha}}_{m'},$$

$$\overset{+}{\boldsymbol{\alpha}}_{mm'} = \hat{\boldsymbol{\alpha}}_m + \hat{\boldsymbol{\alpha}}_{m'},$$

$$\mathbf{b}_{mm'} = \mathrm{B}_{mm'}(2\pi)^2 \mathfrak{L}^{-2}(\boldsymbol{\beta}_m \mathbf{u}_m + \boldsymbol{\beta}_{m'} \mathbf{u}_{m'}),$$

$$\boldsymbol{\beta}_{mm'} = \boldsymbol{\beta}_m + \boldsymbol{\beta}_{m'},$$

$$\mathrm{B}_{mm'} = (2\pi)^{-2} \mathfrak{L}^2 \boldsymbol{\beta}_{mm'}^{-1},$$

$$\mathbf{d}_{mm'}^n = \mathrm{D}_{mm'}^n(\mathrm{B}_{mm'}^{-1}\mathbf{b}_{mm'} + \boldsymbol{\lambda}_n^{-1}\boldsymbol{\mu}_n),$$

$$\mathrm{D}_{mm'}^n = (\mathrm{B}_{mm'}^{-1} + \boldsymbol{\lambda}_n^{-1})^{-1},$$

$$\mathbf{w}_{mm'}^n = \mathbf{b}_{mm'} - \boldsymbol{\mu}_n,$$

$$\mathrm{W}_{mm'}^n = \mathrm{B}_{mm'} + \boldsymbol{\lambda}_n,$$

$$\mathbf{u}_{mm'} = \mathbf{u}_m - \mathbf{u}_{m'},$$

$$\mathbf{U}_{mm'} = (2\pi)^{-2} \mathfrak{L}^2 (\boldsymbol{\beta}_m^{-1} + \boldsymbol{\beta}_{m'}^{-1}),$$

$$\mathrm{Z}_{mm'}^n = \frac{1}{\sqrt{|\mathrm{W}_{mm'}^n||\mathbf{U}_{mm'}|}} e^{-\frac{1}{2}(\mathbf{w}_{mm'}^n{}^T \mathrm{W}_{mm'}^n{}^{-1}\mathbf{w}_{mm'}^n + \mathbf{u}_{mm'}^T \mathbf{U}_{mm'}^{-1}\mathbf{u}_{mm'})},$$

$$\Sigma_{mm'}^2 = \sigma_{\mathrm{power}}^2 \prod_{q=1}^Q \left( \frac{l_q^2}{\sqrt{\beta_{m_q}\beta_{m'_q}}} \right) (M)^{-1},$$

and

$$(\Psi_2)_{mm}^n = \mathbf{E}_{q_{\mathbf{z}_m} q_{\mathbf{h}_n}} \left[ \Phi_{nm}^T \Phi_{nm} \right]$$

$$\overset{\mathrm{B.1}}{=} \mathbf{E}_{q_{\mathbf{h}_n}} \left[ 2\sigma_{\mathrm{power}}^2 (M)^{-1} (\frac{1}{2} + \frac{1}{2} e^{-2\bar{\mathbf{h}}_{nm}^T \boldsymbol{\beta}_m \bar{\mathbf{h}}_{nm}} \cos(2(\hat{\boldsymbol{\alpha}}_m^T(\mathbf{h}_n - \mathbf{u}_m) + \mathrm{b}_m))) \right]$$

$$\overset{\mathrm{B.2}}{=} \sigma_{\mathrm{power}}^2 (M)^{-1} \left( 1 + \tilde{\Sigma}_m^2 \tilde{\mathrm{Z}}_{nm} e^{-2\hat{\boldsymbol{\alpha}}_m^T \tilde{\mathrm{C}}_{nm} \hat{\boldsymbol{\alpha}}_m} \cos(2(\hat{\boldsymbol{\alpha}}_m^T(\mathbf{c}_{nm} - \mathbf{u}_m) + \mathrm{b}_m))) \right),$$

for $m, m' = 1, \ldots, M$, $m = m'$, with

$$\tilde{\Sigma}_m^2 = \sqrt{\prod_{q=1}^Q \left( \frac{l_q^2}{\beta_{m_q}} \right)} 2^{-Q}.$$

### 6.3.2 DRGP-SS(-IP), THE STATISTICS $\Psi_1$, $\Psi_2$

For the versions DRGP-SS(-IP) the statistics are

$$(\Psi_1)_{nm} = \mathbf{E}_{q_{\mathbf{h}_n}} \left[ \Phi_{nm} \right]$$

$$= \mathbf{E}_{q_{\mathbf{h}_n}} \left[ \sqrt{2\sigma_{\mathrm{power}}^2(M)^{-1}} \cos(\hat{\mathbf{z}}_m^T(\mathbf{h}_n - \mathbf{u}_m) + \mathrm{b}_m) \right]$$

$$\overset{\mathrm{B.2}}{=} \sqrt{2\sigma_{\mathrm{power}}^2(M)^{-1}} e^{-\frac{1}{2}\hat{\mathbf{z}}_m^T \boldsymbol{\lambda}_n \hat{\mathbf{z}}_m} \cos(\hat{\mathbf{z}}_m^T(\boldsymbol{\mu}_n - \mathbf{u}_m) + \mathrm{b}_m),$$

for $m = 1, \ldots, M$, $n = 1, \ldots, N$ with

$$\hat{\mathbf{z}}_m = 2\pi(\mathfrak{L}^{-1}\mathbf{z}_m + \mathbf{p}),$$

and

$\Psi_2 = \sum_{n=1}^{N} (\Psi_2)^n$, where

$$
\begin{aligned}
(\Psi_2)_{mm'}^n &= \mathbf{E}_{q_{\mathbf{h}_n}} \left[ \Phi_{nm}^T \Phi_{nm'} \right] \\
&\overset{\text{B.1}}{=} \mathbf{E}_{q_{\mathbf{h}_n}} \left[ 2\sigma_{\text{power}}^2 (M)^{-1} \cos(\hat{\mathbf{z}}_m^T(\mathbf{h}_n - \mathbf{u}_m) + \mathbf{b}_m) \cos(\hat{\mathbf{z}}_{m'}^T(\mathbf{h}_n - \mathbf{u}_{m'}) + \mathbf{b}_{m'}) \right] \\
&= \sigma_{\text{power}}^2 (M)^{-1} \left( e^{-\frac{1}{2}\bar{\mathbf{z}}_{mm'}^T \boldsymbol{\lambda}_n \bar{\mathbf{z}}_{mm'}} \cos(\bar{\mathbf{z}}_{mm'}^T \boldsymbol{\mu}_n - \bar{\rho}_{mm'} + \bar{\mathbf{b}}_{mm'}) \right. \\
&\quad \left. + e^{-\frac{1}{2}\overset{+}{\mathbf{z}}{}_{mm'}^T \boldsymbol{\lambda}_n \overset{+}{\mathbf{z}}{}_{mm'}} \cos(\overset{+}{\mathbf{z}}{}_{mm'}^T \boldsymbol{\mu}_n - \overset{+}{\rho}_{mm'} + \overset{+}{\mathbf{b}}_{mm'}) \right),
\end{aligned}
$$

for $m, m' = 1, \ldots, M$ with

$$
\begin{aligned}
\bar{\rho}_{mm'} &= \hat{\mathbf{z}}_m^T \mathbf{u}_m - \hat{\mathbf{z}}_{m'}^T \mathbf{u}_{m'}, \\
\overset{+}{\rho}_{mm'} &= \hat{\mathbf{z}}_m^T \mathbf{u}_m + \hat{\mathbf{z}}_{m'}^T \mathbf{u}_{m'}, \\
\bar{\mathbf{z}}_{mm'} &= \hat{\mathbf{z}}_m - \hat{\mathbf{z}}_{m'}, \\
\overset{+}{\mathbf{z}}_{mm'} &= \hat{\mathbf{z}}_m + \hat{\mathbf{z}}_{m'},
\end{aligned}
$$

for the other variables, see the defined variables in the DRGP-VSS case.

### 6.3.3 DRGP-(V)SS-IP, THE STATISTICS $\Psi_{reg}$ AND $\Psi_0$

For $\Psi_{reg} = \mathbf{E}_{q_H} [K_{MN} K_{NM}]$ see $\Psi_2$ in Section 6.3.1 but setting $b_m = 0$, $\alpha_m = 0$ and $\beta_m = 1$ for all $m = 1, \ldots, M$.

$\Psi_0$ naturally is given by $\Psi_0 = \text{tr}\left(\mathbf{E}_{q_H}[K_{NN}]\right) = N\sigma_{\text{power}}^2$ because of the chosen SE covariance function.

### 6.3.4   DRGP-(V)SS(-IP), Lower Bounds

In this section we derive the different variational lower bounds for our models DRGP-(V)SS(-IP). We first show the bound $\mathcal{L}_{\text{(V)SS}}^{\text{REVARB}}$ without optimal variational distribution for $a^{(l)}$. Then the bounds $\mathcal{L}_{\text{(V)SS-opt}}^{\text{REVARB}}$ with optimal variational distribution for $a^{(l)}$ follows, as well as the $\mathcal{L}_{\text{(V)SS-IP}}^{\text{REVARB}}$ case.
We use the simplified notation $d\mathbf{A}d\mathbf{Z}d\mathbf{H} = da^{(1)}\ldots da^{(L+1)}d\mathbf{Z}^{(1)}\ldots d\mathbf{Z}^{(L+1)}dh^{(1)}\ldots dh^{(L)}$.

$$\log(p(\mathbf{y}_{H_{\mathbf{x}}+1:}|\mathbf{X}))$$

$$= \log\left(\int p\left(\mathbf{y}_{H_{\mathbf{x}}+1:}, \left[a^{(l)}, \mathbf{Z}^{(l)}, \mathbf{h}^{(l)}, \mathbf{U}^{(l)}\right]_{l=1}^{L+1} |\mathbf{X}\right) d\mathbf{A}d\mathbf{Z}d\mathbf{H}\right)$$

$$= \log\left(\int \frac{Q_{\text{REVARB}}}{Q_{\text{REVARB}}} p\left(\mathbf{y}_{H_{\mathbf{x}}+1:}, a^{(L+1)}, \mathbf{Z}^{(L+1)}, \mathbf{U}^{(L+1)}, \left[a^{(l)}, \mathbf{Z}^{(l)}, \mathbf{h}^{(l)}, \mathbf{U}^{(l)}\right]_{l=1}^{L} |\mathbf{X}\right) d\mathbf{A}d\mathbf{Z}d\mathbf{H}\right)$$

$$\overset{\text{JI}}{\geq} \int Q_{\text{REVARB}} \log\left(\frac{p\left(\mathbf{y}_{H_{\mathbf{x}}+1:}, a^{(L+1)}, \mathbf{Z}^{(L+1)}, \mathbf{U}^{(L+1)}, \left[a^{(l)}, \mathbf{Z}^{(l)}, \mathbf{h}^{(l)}, \mathbf{U}^{(l)}\right]_{l=1}^{L} |\mathbf{X}\right)}{Q_{\text{REVARB}}}\right) d\mathbf{A}d\mathbf{Z}d\mathbf{H}$$

$$= \sum_{l=1}^{L+1} \int q(\mathbf{a}^{(l)})q(\mathbf{Z}^{(l)})q(\mathbf{h}^{(l)})p(\mathbf{h}_{H_{\mathbf{x}}+1:}^{(l)}|a^{(l)}, \mathbf{Z}^{(l)}, \hat{\mathbf{H}}^{(l)}, \mathbf{U}^{(l)})da^{(l)}d\mathbf{Z}^{(l)}dh^{(l)}$$

$$- \sum_{l=1}^{L}\left(\frac{\hat{N}}{2} - \sum_{i=1+H_{\mathbf{x}}-H_{\mathrm{h}}}^{N} \frac{\log(2\pi\lambda_i^{(l)})}{2} + \sum_{i=1+H_{\mathbf{x}}-H_{\mathrm{h}}}^{H_{\mathrm{h}}} \frac{\log(2\pi)}{2} + \frac{\left(\lambda_i^{(l)} + \left(\mu_i^{(l)}\right)^2\right)}{2}\right)$$

$$- \sum_{l=1}^{L+1} \mathbf{KL}(q_{a^{(l)}}||p_{a^{(l)}}) - \mathbf{KL}(q_{\mathbf{Z}^{(l)}}||p_{\mathbf{Z}^{(l)}})$$

$$= -\frac{\hat{N}}{2}\sum_{l=1}^{L+1}\log(2\pi(\sigma_{\text{noise}}^{(l)})^2) - \frac{\mathbf{y}_{H_{\mathbf{x}}+1:}^{T}\mathbf{y}_{H_{\mathbf{x}}+1:}}{2\left(\sigma_{\text{noise}}^{(L+1)}\right)^2}$$

$$+ \frac{\mathbf{y}_{H_{\mathbf{x}}+1:}^{T}\Psi_1^{(L+1)}\mathbf{m}^{(L+1)}}{\left(\sigma_{\text{noise}}^{(L+1)}\right)^2} - \frac{\text{tr}\left(\Psi_2^{(L+1)}(\mathbf{s}^{(L+1)} + \mathbf{m}^{(L+1)}(\mathbf{m}^{(L+1)})^T)\right)}{2\left(\sigma_{\text{noise}}^{(L+1)}\right)^2}$$

$$+ \sum_{l=1}^{L}\left(-\frac{1}{2\left(\sigma_{\text{noise}}^{(l)}\right)^2}\left(\left(\sum_{i=H_{\mathbf{x}}+1}^{N}\lambda_i^{(l)}\right) + (\mu^{(l)})_{H_{\mathbf{x}}+1:}^{T}\mu_{H_{\mathbf{x}}+1:}^{(l)}\right)\right.$$

$$+ \frac{\left((\mu_{H_{\mathbf{x}}+1:}^{(l)})^T\Psi_1^{(l)}\mathbf{m}^{(l)}\right)}{\left(\sigma_{\text{noise}}^{(l)}\right)^2} - \frac{\text{tr}\left(\Psi_2^{(l)}(\mathbf{s}^{(l)} + \mathbf{m}^{(l)}(\mathbf{m}^{(l)})^T)\right)}{2\left(\sigma_{\text{noise}}^{(l)}\right)^2}$$

$$\left. -\frac{\hat{N}}{2} + \sum_{i=1+H_{\mathbf{x}}-H_{\mathrm{h}}}^{N}\frac{\log(2\pi\lambda_i^{(l)})}{2} - \sum_{i=1+H_{\mathbf{x}}-H_{\mathrm{h}}}^{H_{\mathrm{h}}}\frac{\log(2\pi)}{2} - \frac{\left(\lambda_i^{(l)} + \left(\mu_i^{(l)}\right)^2\right)}{2}\right)$$

$$- \sum_{l=1}^{L+1}\mathbf{KL}(q_{a^{(l)}}||p_{a^{(l)}}) - \mathbf{KL}(q_{\mathbf{Z}^{(l)}}||p_{\mathbf{Z}^{(l)}})$$

$$\overset{\text{def}}{=} \mathcal{L}_{\text{VSS}}^{\text{REVARB}},$$

where we derive $\mathcal{L}_{\text{SS}}^{\text{REVARB}}$ by being not variational over $\mathbf{Z}^{(l)}$. Using the optimal distribution for $a^{(l)} \sim \mathcal{N}((A^{(l)})^{-1}(\Psi_1^{(l)})^T\mu_{H_{\mathbf{x}}+1:}^{(l)}, (\sigma_{\text{noise}}^{(l)})^2(A^{(l)})^{-1})$, with $A^{(l)} = \Psi_2^{(l)} + (\sigma_{\text{noise}}^{(l)})^2 I_M$, respective $\mathbf{y}_{H_{\mathbf{x}}+1:}$

for $L + 1$ we obtain

$$
\begin{aligned}
\geq & -\frac{\hat{N} - M}{2} \sum_{l=1}^{L+1} \log\left(\left(\sigma_{\text{noise}}^{(l)}\right)^2\right) - \frac{\mathbf{y}_{H_{\mathbf{x}}+1:}^T \mathbf{y}_{H_{\mathbf{x}}+1:}}{2\left(\sigma_{\text{noise}}^{(\text{L}+1)}\right)^2} \\
& + \frac{\mathbf{y}_{H_{\mathbf{x}}+1:}^T \Psi_1^{(\text{L}+1)} (A^{(L+1)})^{-1} (\Psi_1^{(\text{L}+1)})^T \mathbf{y}_{H_{\mathbf{x}}+1:}}{2\left(\sigma_{\text{noise}}^{(\text{L}+1)}\right)^2} + \frac{\log(|(A^{(L+1)})^{-1}|)}{2} \\
& + \sum_{l=1}^{L} \left( -\frac{1}{2\left(\sigma_{\text{noise}}^{(l)}\right)^2} \left(\left(\sum_{i=H_{\mathbf{x}}+1}^{N} \lambda_i^{(l)}\right) + (\boldsymbol{\mu}_{H_{\mathbf{x}}+1:}^{(l)})^T \boldsymbol{\mu}_{H_{\mathbf{x}}+1:}^{(l)}\right) \right. \\
& + \frac{(\boldsymbol{\mu}_{H_{\mathbf{x}}+1:}^{(l)})^T \Psi_1^{(l)} (A^{(l)})^{-1} (\Psi_1^{(l)})^T \boldsymbol{\mu}_{H_{\mathbf{x}}+1:}^{(l)}}{2\left(\sigma_{\text{noise}}^{(l)}\right)^2} + \frac{\log(|(A^{(l)})^{-1}|)}{2} \\
& \left. -\frac{\hat{N}}{2} + \sum_{i=1+H_{\mathbf{x}}-H_{\text{h}}}^{N} \frac{\log(2\pi\lambda_i^{(l)})}{2} - \sum_{i=1+H_{\mathbf{x}}-H_{\text{h}}}^{H_{\text{h}}} \frac{\log(2\pi)}{2} - \frac{\left(\lambda_i^{(l)} + \left(\mu_i^{(l)}\right)^2\right)}{2} \right) \\
& - \frac{(L+1)\hat{N}}{2} \log(2\pi) - \sum_{l=1}^{L+1} \mathbf{KL}(q_{\boldsymbol{Z}^{(l)}} || p_{\boldsymbol{Z}^{(l)}}) \\
& \overset{\text{def}}{=} \mathcal{L}_{\text{VSS-opt}}^{\text{REVARB}},
\end{aligned}
$$

where we derive $\mathcal{L}_{\text{SS-opt}}^{\text{REVARB}}$ by being not variational over $\boldsymbol{Z}^{(l)}$.

The IP regularization case with $A^{(l)} = \Psi_2^{(l)} + (\sigma_{\text{noise}}^{(l)})^2 K_{MM}^{(l)}$ and $Q_{\text{REVARB}}$, $P_{\text{REVARB}}$ just defined with the priors and variational distributions for $\mathbf{Z}^{(l)}$ and $\mathbf{H}^{(l)}$, is given by:

$$\log(p(\mathbf{y}_{H_{\mathbf{x}}+1:}|\mathbf{X}))$$
$$= \dots \text{see Titsias \& Lawrence (2010) until Equation 14,}$$
$$\stackrel{\text{JI}}{\geq} \mathbf{E}_{p_{\mathbf{a}^{(l)}|\mathbf{U}^{(l)}}}[\sum_{l=1}^{L+1} \exp(\langle \log(\mathcal{N}(\mathbf{h}_{H_{\mathbf{x}}+1:}^{(l)}|\Phi^{(l)}(K_{MM}^{(l)})^{-1}\mathbf{a}^{(l)}, (\sigma_{\text{noise}}^{(l)})^2 I_N)))\rangle_{Q_{\text{REVARB}}})]$$
$$- \left(2\left(\sigma_{\text{noise}}^{(l)}\right)^2\right)^{-1} \sum_{l=1}^{L+1} \Psi_0^{(l)} - \text{tr}((K_{MM}^{(l)})^{-1}\Psi_{reg}^{(l)}) - \mathbf{KL}(Q_{\text{REVARB}}||P_{\text{REVARB}})$$
$$= -\frac{\hat{N}-M}{2} \sum_{l=1}^{L+1} \log\left(\left(\sigma_{\text{noise}}^{(l)}\right)^2\right) - \frac{\mathbf{y}_{H_{\mathbf{x}}+1:}^T \mathbf{y}_{H_{\mathbf{x}}+1:}}{2\left(\sigma_{\text{noise}}^{(L+1)}\right)^2} + \frac{\log(|K_{MM}^{(L+1)}|)}{2}$$
$$+ \frac{\mathbf{y}_{H_{\mathbf{x}}+1:}^T \Psi_1^{(L+1)}(A^{(L+1)})^{-1}(\Psi_1^{(L+1)})^T \mathbf{y}_{H_{\mathbf{x}}+1:}}{2\left(\sigma_{\text{noise}}^{(L+1)}\right)^2} + \frac{\log(|(A^{(L+1)})^{-1}|)}{2}$$
$$- \frac{\Psi_0^{(L+1)} + \text{tr}((K_{MM}^{(L+1)})^{-1}\Psi_{reg}^{(L+1)})}{2\left(\sigma_{\text{noise}}^{(L+1)}\right)^2}$$
$$- \sum_{l=1}^{L} \left(-\frac{1}{2\left(\sigma_{\text{noise}}^{(l)}\right)^2}\left(\left(\sum_{i=H_{\mathbf{x}}+1}^{N} \lambda_i^{(l)}\right) + (\boldsymbol{\mu}_{H_{\mathbf{x}}+1:}^{(l)})^T \boldsymbol{\mu}_{H_{\mathbf{x}}+1:}^{(l)}\right) + \frac{\log(|K_{MM}^{(l)}|)}{2}\right.$$
$$+ \frac{(\boldsymbol{\mu}_{H_{\mathbf{x}}+1:}^{(l)})^T \Psi_1^{(l)}(A^{(l)})^{-1}(\Psi_1^{(l)})^T \boldsymbol{\mu}_{H_{\mathbf{x}}+1:}^{(l)}}{2\left(\sigma_{\text{noise}}^{(l)}\right)^2} + \frac{\log(|(A^{(l)})^{-1}|)}{2}$$
$$- \frac{\Psi_0^{(l)} + \text{tr}((K_{MM}^{(l)})^{-1}\Psi_{reg}^{(l)})}{2\left(\sigma_{\text{noise}}^{(l)}\right)^2} - \frac{\hat{N}}{2} + \sum_{i=1+H_{\mathbf{x}}-H_{\text{h}}}^{N} \frac{\log(2\pi\lambda_i^{(l)})}{2}$$
$$\left. - \sum_{i=1+H_{\mathbf{x}}-H_{\text{h}}}^{H_{\text{h}}} \frac{\log(2\pi)}{2} - \frac{\left(\lambda_i^{(l)} + \left(\mu_i^{(l)}\right)^2\right)}{2}\right) - \frac{(L+1)\hat{N}}{2}\log(2\pi) - \sum_{l=1}^{L+1} \mathbf{KL}(q_{\mathbf{Z}^{(l)}}||p_{\mathbf{Z}^{(l)}})$$
$$\stackrel{\text{def}}{=} \mathcal{L}_{\text{VSS-IP}}^{\text{REVARB}},$$

where again we derive $\mathcal{L}_{\text{SS-IP}}^{\text{REVARB}}$ by being not variational over $\mathbf{Z}^{(l)}$.

### 6.3.5 PREDICTIONS

Predictions for each layer $l$ and new $\hat{\mathbf{h}}_*^{(l)}$ are performed in the simple DRGP-(V)SS case with

$$\mathbf{E}_{q_{f^{*(l)}}}\left[f^{*(l)}\right] = \Psi_{1*}^{(l)}\mathbf{m}^{(l)},$$

$$\mathbf{V}_{q_{f^{*(l)}}}\left[f^{*(l)}\right] = \left(\mathbf{m}^{(l)}\right)^T\left(\Psi_{2*}^{(l)} - \left(\Psi_{1*}^{(l)}\right)^T\Psi_{1*}^{(l)}\right)\mathbf{m}^{(l)} + \mathrm{tr}\left(\mathbf{s}^{(l)}\Psi_{2*}^{(l)}\right),$$

where for the optimal distribution case for $\boldsymbol{a}^{(l)}$ we have with $A^{(l)} = \Psi_2^{(l)} + (\sigma_{\mathrm{noise}}^{(l)})^2 I_M$

$$\mathbf{m}^{(l)} \overset{\mathrm{opt}}{=} \left(A^{(l)}\right)^{-1}\left(\Psi_1^{(l)}\right)^T\boldsymbol{\mu}_{H_\mathbf{x}+1:,}^{(l)}, \quad \mathbf{s}^{(l)} \overset{\mathrm{opt}}{=} (\sigma_{\mathrm{noise}}^{(l)})^2\left(A^{(l)}\right)^{-1},$$

for $1,\ldots,L$, and fully analog for $l = L+1$ by replacing $\boldsymbol{\mu}_{H_\mathbf{x}+1:}^{(l)}$ with $\mathbf{y}_{H_\mathbf{x}+1:}$.

In the DRGP-(V)SS-IP case we make predictions for each layer $l$ and new $\hat{\mathbf{h}}_*^{(l)}$ through

$$\mathbf{E}_{q_{f^{*(l)}}}\left[f^{*(l)}\right] = \Psi_{1*}^{(l)}\boldsymbol{\Lambda}^{(l)},$$

$$\mathbf{V}_{q_{f^{*(l)}}}\left[f^{*(l)}\right] = \left(\boldsymbol{\Lambda}^{(l)}\right)^T\left(\Psi_{2*}^{(l)} - \left(\Psi_{1*}^{(l)}\right)^T\Psi_{1*}^{(l)}\right)\boldsymbol{\Lambda}^{(l)}$$

$$+ \Psi_0 - \mathrm{tr}\left((K_{MM}^{(l)})^{-1}\Psi_{reg*}^{(l)} - (\sigma_{\mathrm{noise}}^{(l)})^2\left(A^{(l)}\right)^{-1}\Psi_{2*}^{(l)}\right),$$

where $A^{(l)} = \Psi_2^{(l)} + (\sigma_{\mathrm{noise}}^{(l)})^2 K_{MM}^{(l)}$ and

$$\boldsymbol{\Lambda}^{(l)} \overset{\mathrm{opt}}{=} \left(A^{(l)}\right)^{-1}\left(\Psi_1^{(l)}\right)^T\boldsymbol{\mu}_{H_\mathbf{x}+1:,}^{(l)}$$

for $1,\ldots,L$, and fully analog for $l = L+1$ by replacing $\boldsymbol{\mu}_{H_\mathbf{x}+1:}^{(l)}$ with $\mathbf{y}_{H_\mathbf{x}+1:}$.

Prediction for a new input in all cases has time complexity $\mathcal{O}((L+1)M^3)$, which comes from the iterative prediction through all GP layers and the calculation of the statistics, see Appendix 6.3.6.

### 6.3.6 DISTRIBUTED VARIATIONAL INFERENCE FOR DRGP-(V)SS(-IP)

We refer to (Gal et al., 2014), Equation (4.3), for a comparison.

Calculating the *optimal* REVARB-(V)SS and REVARB-(V)SS-IP requires $\mathcal{O}(NM^2Q_{\max}(L+1))$, where $Q_{\max} = \max\limits_{l=1\dots,L+1} Q^{(l)}$, $Q^{(l)} \overset{\text{def}}{=} \dim(\hat{\mathbf{h}}_i^{(l)})$ and $\hat{\mathbf{h}}_i^{(l)}$ is coming from the Equations in (13) for a fixed chosen $i$ and $l = 1, \dots, L+1$. In this section we show how we can reduce the complexity of inference in the REVARB-(V)SS(-IP) setting with distributed inference to $\mathcal{O}(M^3)$ if the number of cores scales suitably with the number of training-data. We show this for $\mathcal{L}_{\text{(V)SS-IP}}^{\text{REVARB}}$, because this is the most complex bound, and all other bounds reduce to special cases of this.

$\mathcal{L}_{\text{(V)SS-IP}}^{\text{REVARB}}$ in Appendix 6.3.4, separated for each hidden layer and the output layer ($\boldsymbol{\mu}_{H_\mathbf{x}+1:}$ and $\boldsymbol{\lambda}_{H_\mathbf{x}+1:}$ replaced by $\mathbf{y}_{H_\mathbf{x}+1:}$), can be written as $\mathcal{L}_{\text{(V)SS-IP}}^{\text{REVARB}} = \sum\limits_{l=1}^{L+1} B_l$, where we have the KL terms

$$\mathbf{KL}(q_{\mathbf{Z}^{(l)}}||p_{\mathbf{Z}^{(l)}}) = \frac{1}{2}\sum_{m=1}^{M} \text{tr}\left(\boldsymbol{\beta}_m^{(l)} + \boldsymbol{\alpha}_m^{(l)}(\boldsymbol{\alpha}_m^{(l)})^T\right) - \frac{1}{2}\sum_{m=1}^{M} \log(|\boldsymbol{\beta}_m^{(l)}|) - \frac{MQ^{(l)}}{2},$$

for $l = 1, \dots, L+1$, the terms

$$-\frac{\hat{N}}{2} + \sum_{i=1+H_\mathbf{x}-H_\text{h}}^{N} \frac{\log(2\pi\lambda_i^{(l)})}{2} - \sum_{i=1+H_\mathbf{x}-H_\text{h}}^{H_\text{h}} \frac{\log(2\pi)}{2} - \frac{\left(\lambda_i^{(l)} + \left(\mu_i^{(l)}\right)^2\right)}{2},$$

for $l = 1, \dots, L$ and

$$B_l = -\frac{\hat{N}-M}{2}\log\left(\left(\sigma_{\text{noise}}^{(l)}\right)^2\right) - \frac{\left(\sum_{i=H_\mathbf{x}+1}^{N}\lambda_i^{(l)}\right) + \boldsymbol{\mu}_{H_\mathbf{x}+1:}^T\boldsymbol{\mu}_{H_\mathbf{x}+1:}}{2\left(\sigma_{\text{noise}}^{(l)}\right)^2} + \frac{\log(|K_{MM}^{(L+1)}|)}{2}$$

$$+ \frac{\boldsymbol{\mu}_{H_\mathbf{x}+1:}^T\Psi_1^{(l)}\left(A^{(l)}\right)^{-1}(\Psi_1^{(l)})^T\boldsymbol{\mu}_{H_\mathbf{x}+1:}}{2\left(\sigma_{\text{noise}}^{(l)}\right)^2} + \frac{\log(|(A^{(l)})^{-1}|)}{2} - \frac{\Psi_0^{(l)} + \text{tr}((K_{MM}^{(l)})^{-1}\Psi_{reg}^{(l)})}{2\left(\sigma_{\text{noise}}^{(l)}\right)^2}$$

$$- \frac{\hat{N}}{2}\log(2\pi),$$

for all $l = 1, \dots, L+1$ and which can be separated further into $B_l = \sum_{i=H_\mathbf{x}+1}^{N} B_{li}$, a sum of $\hat{N} = N - H_\mathbf{x}$ independent terms, extracting $\frac{\log(|(A^{(l)})^{-1}|)}{2}$, $\frac{\log(|K_{MM}^{(l)}|)}{2}$ and $\frac{\text{tr}((K_{MM}^{(l)})^{-1}\Psi_{reg}^{(l)})}{2\left(\sigma_{\text{noise}}^{(l)}\right)^2}$, by

$$B_{li} = -\frac{1-\frac{M}{\hat{N}}}{2}\log\left(\left(\sigma_{\text{noise}}^{(l)}\right)^2\right) - \frac{\left(\sigma_{\text{power}}^{(l)}\right)^2 + \lambda_i^{(l)} + \mu_i^{(l)}\mu_i^{(l)}}{2\left(\sigma_{\text{noise}}^{(l)}\right)^2}$$

$$+ \frac{\mu_i^{(l)}(\Psi_1^{(l)})_{\cdot i}(A^{(l)})^{-1}(\Psi_1^{(l)})_{\cdot i}^T\mu_i^{(l)}}{2\left(\sigma_{\text{noise}}^{(l)}\right)^2} + \frac{\log(2\pi\lambda_i^{(l)})}{2} - \frac{\log(2\pi)}{2},$$

where $(\Psi_1^{(l)})_{\cdot i}$ means, taking the $i$-th column of $\Psi_1^{(l)}$ for $l = 1, \dots, L+1$, $i = H_\mathbf{x}+1 \dots, N$. We further inspect

$$A^{(l)} = \Psi_2^{(l)} + (\sigma_{\text{noise}}^{(l)})^2 K_{MM}^{(l)} = \sum_{i=H_\mathbf{x}+1}^{N}\left(\Psi_2^i\right)^{(l)} + (\sigma_{\text{noise}}^{(l)})^2 K_{MM}^{(l)},$$

and

$$\Psi_{reg}^{(l)} = \sum_{i=H_\mathbf{x}+1}^{N}\left(\Psi_{reg}^i\right)^{(l)}.$$

These terms and the sums of $\Psi_2^{(l)}$ and $\Psi_{reg}^{(l)}$ can be computed on different cores in parallel without communication. Only the $3(L+1)$ inversions and determinants of $A^{(l)}$ and $K_{MM}^{(l)}$ now are responsible for the complexity, which can also be computed on $3(L+1)$ cores. Summing this bound over $i$ and $l$, we obtain the total complexity of $\mathcal{O}(M^3)$ per single iteration with $\hat{N}(L+1)$ cores.

