# OpenReview forum: "Deep Recurrent Gaussian Process with Variational Sparse Spectrum Approximation"
_ICLR.cc/2019/Conference_

### Official Review · AnonReviewer2 · 2018-11-02
**DEEP RECURRENT GAUSSIAN PROCESS WITH VARIATIONAL SPARSE SPECTRUM APPROXIMATION**

**Rating:** 7
**Confidence:** 2

**Review:**

Overall Score: 7/10.
Confidence Score: 7/10.

Detailed Comments: This paper introduces various Deep Recurrent Gaussian Process (DRGP) models based on the Sparse Spectrum Gaussian Process (SSGP) models and the Variational Sparse Spectrum Gaussian Process (VSSGP) models. This is a good paper and proposed models are very sound so I recommend for acceptance although as main weakness I can say that is very technical so it can be difficult to follow. Adding more intuitive ideas, motivation and maybe a figure for each step would be a solution. Apart from that it is a really good paper, congratulations.

Related to: RNN models and Sparse Nystrom approximation.

Strengths: Models are very sound, solutions are solid, the proposed methodology is correct and the empirical results and experiments are valid and properly done.

Weaknesses: It is too difficult to follow and it is written in an extreme technical way. More intuitions and a proper motivation both in the abstract and introduction may be put in order to make the paper easier to read and, hence, more used by researchers and data scientists.

Does this submission add value to the ICLR community? : Yes it does, the experiments show the efficiency of the proposed methods in some scenarios and are valid methodologies.

Quality:
Is this submission technically sound?: Yes it is.
Are claims well supported by theoretical analysis or experimental results?: Experimental results prove empirically the methods and appendixes show the analysis performed in a clear and elegant way.
Is this a complete piece of work or work in progress?: Complete piece of work.
Are the authors careful and honest about evaluating both the strengths and weaknesses of their work?: Yes, and I would enfatize that I have liked that some experiments are won by other methods such as GP-LSTM, they are very honest.

Clarity:
Is the submission clearly written?: Yes, but it is difficult for newcomers due to the reasons that I have stated before.
Is it well organized?: Yes it is.
Does it adequately inform the reader?: Yes it is.

Originality:
Are the tasks or methods new?: Yes, they are sound.
Is the work a novel combination of well-known techniques?: Yes it is.
Is it clear how this work differs from previous contributions?: Yes.
Is related work adequately cited?: Yes, being a strength of the paper.

Significance:
Are the results important?: I would argue that they are and are a clear alternative to consider in order to solve these problems.
Are others likely to use the ideas or build on them?: If the paper is written in a more friendly way, yes.
Does the submission address a difficult task in a better way than previous work?: Yes I think.
Does it advance the state of the art in a demonstrable way?: Yes, empirically.

Arguments for acceptance: Models are very sound, solutions are solid, the proposed methodology is correct and the empirical results and experiments are valid and properly done

Arguments against acceptance: Clarity of the paper.

Minor issues and typos:
-> (V)SS not defined before being used.
-> Abstract should be rewritten adding a motivation and focusing more on the problems being solved and less in the details of the solutions.
-> Recurrent indexes that go backwards (i) of Eq. 1. should be explained why are going backwards before being used like that. Newcomers may be confused.
-> Section 2 writing style lacks a bit of cohesion, relating the paragraphs may be a solution.
-> Q is not defined in section 3.1 paragraph 1.
-> A valid covariance function must produce a PSD matrix, put that in section 3.1.
-> I do not see how U marginalizes in Eq. 7, kind of confused about that, I think that it should be p(y|X,U).
-> Section 3.4 statistics should be explained.

Reading thread and authors response rebuttal decision:
=================================================

I consider that the authors have perfomed a good rebuttal and reading the other messages and the authors response I also consider that my issue with clarity is solved. Hence, I upgrade my score to 7 and recommend the paper for publication.

---

> ### Author Response · Authors · 2018-11-25
> **Response to AnonReviewer2**
>
> We greatly appreciate your insightful feedback. We would like to respond to your comments and explain how we improved the manuscript.
>
> First of all, we recognized that you changed your rating from
>
> <<Rating: 7, Confidence: 3>>          to           <<Rating: 5: Confidence: 2>> ,
>
> without any further explanations and without waiting for our responses.
> We do not know, what your concerns are, but we hope, we can address these with our answers now.
>
> <<Arguments against acceptance: Clarity of the paper.
> -> Section 2 writing style lacks a bit of cohesion, relating the paragraphs may be a solution.>>
>
> Thanks, we addressed this issue as good as possible.
>
> <<-> Abstract should be rewritten adding a motivation and focusing more on the problems being solved and less in the details of the solutions. >>
>
> Thank you for the advice. We reformulated the abstract. We added:
> “Our approach can deal with a larger class of covariance functions than the RGP, because its spectral nature allows variational integration in all stationary cases.”
>
> “For the DRGP extension of these combined approximations and the simple (V)SS approximations an optimal variational distribution exists.”
>
> And deleted
>
> “This case naturally collapses to a tractable expression by calculating the integrals. For the simple
> extension of the (V)SS approximation an optimal variational distribution exists. “
>
>  “Training is realized through optimizing the variational lower bounds.”
>
> <<-> Recurrent indexes that go backwards (i) of Eq. 1. should be explained why are going backwards before being used like that. Newcomers may be confused.
> Minor issues and typos: -> (V)SS not defined before being used.
> -> Q is not defined in section 3.1 paragraph 1.
> -> A valid covariance function must produce a PSD matrix, put that in section 3.1. >>
>
> Thank you for pointing out these issues.
> We see your point with the time horizon and simplified it to H, similar to our experiments.
> We addressed the issue with (V)SS and also defined in the abstract the GP.
> We added in Section 3.1:
> “ …,Q ∈ N the input-dimension,…”
> “Be aware of that a valid covariance function must produce a positive definite matrix K NN
> def= (k(x i ,x j )) N i,j=1 ∈ R N×N , when filling in combinations of data-input points x i , i = 1,...,N.”
>
> <<-> I do not see how U marginalizes in Eq. 7, kind of confused about that, I think that it should be p(y|X,U).>>
>
> You are correct that the LHS depends on U and we have to make this point clear to the reader. Additionally, the LHS depends on further variables like L, p, b. etc. Nevertheless, we decided to highlight U just in p(y|a,Z,U,X) in the integral. On the one hand, our approximations are built on U in the following sections (c.f. Section 3.3.) and on the other hand we want to be notationally conform with Gal (2016), Section 3. Having said this, for clarity we added:
> “…highlighting U just in the integral, to be notationally conform to Gal & Turner (2015), Section 3.”
>
>  <<-> Section 3.4 statistics should be explained.>>
>
> Thank you for the advice. We added:
> “These statistics are essentially the given matrices Φ, Φ T Φ, K_MN K_NM from the beginning, but every input h_i and every spectral point z_m are now replaced by a mean µ_i , α_m and a variance λ_i , β_m resulting in matrices of the same size. The property of positive definiteness is preserved.”
>
> Thank you. Sincerely,
>
> The authors

---

### Official Review · AnonReviewer1 · 2018-11-02
**Combination of known ideas, hard to read**

**Rating:** 5
**Confidence:** 4

**Review:**

This paper proposes deep recurrent GP models based on the existing DRGP framework, two works on sparse spectrum approximation as well as that of inducing points. In these models, uncertainty is propagated by marginalizing out the hidden inputs at every layer.

The authors have combined a series of known ideas in the proposed work. There is a serious lack of discussion or technical insights from the authors for their technical formulations: in particular, what are the non-trivial technical challenges addressed in the proposed work? Furthermore, the authors are quite sloppy in referencing equations and inconsistent in the use of their defined notations and acronyms. I also find it hard to read and understand the main text due to awkward sentence structures.

Have the authors revealed their identity on page 2 of the paper? I quote: "We refer to the report Foll et al. (2017) for a detailed but preliminary formulation of our models and experiments." and "DRGP-(V)SS code available from http://github.com/RomanFoell/DRGP-VSS."



Detailed comments are provided below:

For the first contribution stated by the authors, what are the theoretical and practical implications of the different regularization terms/properties between the lower bounds in equations 10 vs. 8? These are not described in the paper.

Can the authors provide a detailed derivation of DVI for equation 13 as well as for the predictive distributions in Sectio 6.3.5?

Can the authors provide a time complexity analysis of all the tested deep recurrent GPs?


Would the authors' proposed approach be able to extend the framework of Hoang et al. (2017) (see below) that has generalized the SS approximation of Lazaro-Gredilla et al. (2010) and the improved VSS approximation of Gal & Turner (2015)?

Hoang, Q. M.; Hoang, T. N.; and Low, K. H. 2017. A generalized stochastic variational Bayesian hyperparameter learning framework for sparse spectrum Gaussian process regression. In Proc. AAAI, 2007–2014.



Minor issues:
Just below equation 6, equation 9, and throughout the entire paper, the authors need to decide whether to italicize their notations in bold or not.

Equations are not properly referenced in a number of instances.

The authors have used their commas too sparingly, which makes some sentences very hard to parse.

What is the difference between REVARB-(V)SS(-IP), DRGP-(V)SS(-IP), and DRGP-VSS-IP?

Equation 7: LHS should be conditioned on U.
Page 4:  (V)SSGP does not have the same...
Equation 8: q_a and q_Z should be placed next to the expectation.
Page 4: choosen?
Page 5: will makes it possible?
Page 5: DRGP-SSGP, -VSSGP, -SSGP-IP, -VSSG-IP?
Page 5: to simplify notation, we write h^{L+1}_{Hx+1:} = y_{Hx+1:}? Such a notation does not look simplified.

Equation after equation 12: On LHS, should U^(l) be a random variable?

Page 17: Should the expressions begin with >=?

---

> ### Author Response · Authors · 2018-11-25
> **Response to AnonReviewer1**
>
> We greatly appreciate your insightful feedback. We would like to respond to your comments and explain how we improved the manuscript.
>
> <<For the first contribution stated by the authors, what are the theoretical and practical implications of the different regularization terms/properties …>>
>
> Thank you for asking for details. The practical implications are, that the GP is regularized during optimization when optimizating over U. These parameters U have, following Gal & Turner (2015), the same properties as in the Nyström case, but in the lower bound in (8) they are simply used without being linked to the weights a.
> What we can show, is, that we can further marginalize the integral in (7) to a Gaussian with mean 0 and covariance matrix
> K = K_NN + (Φ (K_MM)^-1 Φ^T - K_NM (K_MM)^-1 K_MN) + σ_noise^2.
> We see that we have the true covariance matrix plus a discrepancy of the sparse approximations of the Sparse Spectrum and the Nyström covariance matrix plus noise. Therefore, our approximation can be seen as a trade-off between these two sparse approximations.
> For now we included these insights in Section 3.3.
> We further fixed a typo in the prior assumption and added the noise. We also made a distinction for the IP case in Section 4.2.
>
> <<Can the authors provide a detailed derivation of DVI for equation 13 as well as for the predictive distributions in Section 6.3.5? >>
>
> Please refer to Section 6.3.6. and to the end of Section 6.3.5. for the detailed derivation.
>
> <<Can the authors provide a time complexity analysis of all the tested deep recurrent GPs? >>
>
> Since the different implementations have different evolution states with regard to optimization, we decided against a time complexity analysis, since we think it might be misleading. But we agree, that your question is of interest and should be done in an extra work.
>
> <<Would the authors' proposed approach be able to extend the framework of Hoang et al. (2017) (see below) … >>
>
> Thank you for making us aware of this paper. We do not see any problems. We added a sentence in Section 5.3.
>
> <<Minor issues: Just below equation 6, equation 9, … need to decide whether to italicize their notations in bold or not. >>
>
> Thank you for the advice. We consciously decided to use italic and bold. We have to make this point clear to the reader.  To make this clearer to the reader, we added an explanation in beginning of Section 3. We hope this will meet your concerns. We fixed an issue for the expectation in Section 3.1, write it in bold as in the entire paper and also write f_x instead of f(x) for the random variable at x.
>
> <<Equations are not properly referenced in a number of instances. >>
> <<The authors have used their commas too sparingly, which makes some sentences very hard to parse. >>
> Thanks for your comment. We tried to address these issues as good as possible.
>
> <<What is the difference between REVARB-(V)SS(-IP), DRGP-(V)SS(-IP), and DRGP-VSS-IP?
> Page 5: DRGP-SSGP, -VSSGP, -SSGP-IP, -VSSG-IP>>
>
> REVARB-(V)SS(-IP) is the name of the lower bounds, overall four cases.
> DRGP-(V)SS(-IP) is the name of the corresponding models/methods.
>
> We changed beginning of Section 4. and beginning of Section 4.2. to be more clear.
>
> <<Equation 7: LHS should be conditioned on U. >>
>
> You are correct that the LHS depends on U and we have to make this point clear to the reader. Additionally, the LHS depends on further variables like L, p, b. etc. Nevertheless, we decided to highlight U just in p(y|a,Z,U,X) in the integral. On the one hand, our approximations are built on U in the following sections (c.f. Section 3.3.) and on the other hand we want to be notationally conform with Gal (2016), Section 3. Having said this, for clarity we added:
> “…, highlighting U just in the integral, to be notationally conform to Gal & Turner (2015), Section 3.”
>
> <<Page 4: (V)SSGP does not have the same...
> Equation 8: q_a and q_Z should be placed next to the expectation.
> Page 4: choosen?
> Page 5: will makes it possible? >>
>
> Thank you for pointing out all those issues, we addressed all of them.
>
> <<Page 5: to simplify notation, we write h^{L+1}_{Hx+1:} = y_{Hx+1:}?
> Such a notation does not look simplified. >>
>
> The simplification comes from the special role of the last layer, where h^(L+1) corresponds with y. In order to avoid these notational issues throughout the paper, we introduced the respective notation (see the joint density and Equations 14, 15). Anyhow, we changed the position of the simplification right below the joint density.
>
> <<Equation after equation 12: On LHS, should U^(l) be a random variable? >>
>
> We did not define any prior or variational distribution on these parameters, but the standard notation is, to assume it is. We also write f_X |X, where e.g. over X no distribution is defined.
>
> <<Page 17: Should the expressions begin with >=?>>
>
> Yes, in order to highlight that they are lower bounds.
>
> Thank you. Sincerely,
>
> The authors

---

### Official Review · AnonReviewer3 · 2018-11-03
**A combination of existing sparse-spectrum techniques and deep recurrent Gaussian processes but not properly justified**

**Rating:** 5
**Confidence:** 4

**Review:**

This paper addresses the problem of modeling sequential data based on one of the deep recurrent Gaussian process (DRGP) structures proposed by Mattos et al (2016). This structure acts like a recurrent neural net where every layer is defined as a GP. One of the main limitations of the original method proposed by Mattos et al (2016) is that it is limited to a small set of covariance functions, as the variational expectations over these have to be analytically tractable.

The main contributions of this paper are the use of previously proposed inference, namely (i) the sparse spectrum (SS) of Lazaro-Gredilla et al (2010); its variational improvement by Gal and Turnner (2015) (VSS);  and the inducing-point (IP) framework of Titsias and Lawrence (2010) into the recurrent setting of Mattos et al (2016). Most (if not all) of the technical developments in the paper are straightforward applications of the results in the papers above. Therefore, the technical contribution of the paper is largely incremental. Furthermore, while it is sensible to use random-feature approximation approaches (such as SS and VSS) in GP models, it is very unclear why combining the IP framework with SS approaches makes any sense at all. Indeed, the original IP framework was motivated as a way to deal with the scalability issue in GP models, and the corresponding variational formulation yielded a nice property of an additional regularization term in the variational bound. However, making the prior over a (Equation 9) conditioned on the inducing variables U is rather artificial and lacks any theoretical justification. To elaborate on this, in the IP framework both the latent functions (f in the original paper) and the inducing inputs come from the same GP prior, hence having a joint distribution over these comes naturally. However, in the approach proposed in this paper, a is a simple prior over the weights in a linear-in-the-parameters model, and from my perspective, having a prior conditioned on the inducing variables lacks any theoretical motivation.

The empirical results are a bit of a mixed bag, as the methods proposed beat (by a small margin) the corresponding benchmarks on 6 out of 10 problems. While one would not expect a proposed method to win on all possible problems (no free lunch), it will be good to have some insights into when the proposed methods are expected to be better than their competitors.

While the proposed method is motivated from an uncertainty propagation perspective, only point-error metrics (RMSE) are reported. The paper needs to do a proper evaluation of the full predictive posterior distributions. What is the point of using GPs otherwise?

Other comments:
I recommend the authors use the notation p(v) = … and q(v) = … everywhere rather than v ~ … as the latter may lead to confusion on how the priors and the variational distributions are defined.
It is unnecessary to cite Bishop to explain how one obtains a marginal distribution
Would it be possible to use the work of Cutajar et al (2017), who use random feature expansions for deep GPs,  in the sequential setting? If so, why aren’t the authors comparing to this?
The analysis of Figure 1 needs expanding
What are the performance values obtained with a standard recurrent neural net / LSTM?

---

> ### Author Response · Authors · 2018-11-25
> **Response to AnonReviewer3**
>
> We greatly appreciate your insightful feedback. We would like to respond to your comments and explain how we improved the manuscript.
>
> <<Most (if not all) of the technical developments in the paper are straightforward applications … technical contribution of the paper is largely incremental.>>
>
> You are correct that many of the steps we take are incremental and we face the same difficulties as in Titsias & Lawrence (2010), Gal & Turner (2015) and Mattos et al. (2016). Nevertheless, we think that the combination of these methods is new, as well the integration of the input space for spectral covariance function, which is not straightforward. Also, the combination of (V)SSGP with the regularization property of Titsias & Lawrence (2009);(2010) is not clear from the start. All in all, we hope that the sum of all these steps gives a valuable contribution to the community.
>
> <<Furthermore, while it is sensible to use random-feature approximation approaches (such as SS and VSS), … and from my perspective, having a prior conditioned on the inducing variables lacks any theoretical motivation. >>
>
> These parameters U have, following Gal & Turner (2015), the same properties as in the Nyström case, but in the lower bound in (8) they are simply used without being linked to the weights a.
> What we can show, is, that we can further marginalize the integral in (7) to a Gaussian with mean 0 and covariance matrix
> K = K_NN + (Φ (K_MM)^-1 Φ^T - K_NM (K_MM)^-1 K_MN) + σ_noise^2.
> We see that we have the true covariance matrix plus a discrepancy of the sparse approximations of the Sparse Spectrum and the Nyström covariance matrix plus noise. Therefore, our approximation can be seen as a trade-off between these two sparse approximations.
> For now we included these insights in Section 3.3.
> We further fixed a typo in the prior assumption and added the noise. We also made a distinction for the IP case in Section 4.2.
>
> <<The empirical results are a bit of a mixed bag, …, it will be good to have some insights into when the proposed methods are expected to be better than their competitors. >>
>
> Of course you are correct. Our approach does not beat all corresponding benchmarks. The question, what method should be used in what setting is of great interest and we want to address this topic in future work. Currently we cannot characterize situations, when the new methods are better. Nevertheless, we hope that our work shows the capabilities of our approach and therefore is of interest.
>
> <<While the proposed method is motivated from an uncertainty propagation perspective, … predictive posterior distributions. What is the point of using GPs otherwise? >>
>
> We compared our predictive posterior distribution with the one of Mattos et al. (2016). We implemented his version by ourselves and did not have the same problems with variance predictions as in his paper Mattos et al. (2016) Figure 2, (l) (we therefore think that it might have been an implementation issue).
> All in all we think, that the variance predictions are equally good and therefore we concentrated on the RSME comparison. We added two more sentences in Section 5.2 regarding this.
>
> <<It is unnecessary to cite Bishop to explain how one obtains a marginal distribution. >>
>
> Yes, it is straightforward. We deleted the reference before Equation 7, but we think that it might be of interest for a reader unfamiliar with the topic, and so we kept the reference in Section 3.3.
>
> <<Would it be possible, to use the work of Cutajar et al (2017), … If so, why aren’t the authors comparing to this? >>
>
> Thank you for the input. We added the experiments for this DGP with NARX structure for the first layer to our paper.
>
> <<I recommend the authors use the notation p(v) = … and q(v) = … everywhere rather than v ~ …>>
>
> Thank you for your advice. We agree that the notation p(v)=... is widely used. Nevertheless, we prefer to be mathematically more precise and differentiate between the random variable v (italic) and the realizations/samples v (upright) and therefore we use the notation p_v, where v ~ N(m,v). We agree, that our paper was not very specific in highlighting that point. To make this clearer to the reader, we added an explanation in beginning of Section 3. We hope this will meet your concerns.
>
> <<The analysis of Figure 1 needs expanding. >>
>
> Thank you for the advice. We added the following explanation and hope that we meet your concerns:
> ”We initialize the states with the output training-data for all layers with minor noise (first column) and after training we obtain a trained state (second column).”
>
> <<What are the performance values obtained with a standard recurrent neural net / LSTM? >>
>
> We agree, that a comparison with those methods is of interest and we therefore added the results of Al-Shedivat et al. (2017) and our own results for standard RNN and LSTM. We further added also the results, where we deleted the auto-regressive part in the first layer for GP-LSTM.
>
> Thank you. Sincerely,
>
> The authors

---

### Author Response · Authors · 2018-11-25
**Response to All Reviewer**

We greatly appreciate all your insightful feedback. We would like to respond to your comments in general and explain how we improved the manuscript.
In particular, we included three more references:  Eleftheriadis et al. (2017) for SSM models, (Rahimi & Recht, 2008) for the case of Random Fourier Features, Hoang et al. (2017) for a generalization of the (V)SSGP.
More methods in the experiments are included, the RNN, the LSTM and the DGP-RFF from Cutajar et al. (2016).
We improved the appendix.
Furthermore, we added the detailed derivation of DVI in the appendix.
We further changed the appearance of the density for all expectations, bringing it to the front, for more readability.
There was a discussion about our notation (upright, bold and italic), therefore we added at the beginning of Section 3. an explanation.
In Section 3.3 we explain more in detail how the combination with the IP case makes sense from the theoretical point and hope this will meet your concerns.
In Section 3.4 we explain the upcoming statistics more in detail.
We further improved many language issues, punctuation problems and hope the paper improved its clarity.

Details are explained in the individual responses to each referee report.

Thank you. Sincerely,

The authors.

---

### Author Response · Authors · 2018-11-27
**Response to All Reviewer**

We are sorry for a 'typo' in the lower bound in Equation (15) regarding the expectation:

Actually one has to replace

E_p_l[  sum_l   ...  ]

with

log(  prod_l  E_p_l[  ...  ]  )

We are sorry for this and will fix this in the next upload phase.

Thank you. Sincerely,

The authors.

---

### Meta-Review · Area_Chair1 · 2018-12-13
**New model but presentation and insights need to improve.**

**Confidence:** 4
**Recommendation:** Reject

**Metareview:**

This paper is concerned with combining past approximation methods to obtain a variant of Deep Recurrent GPs. While this variant is new, 2/3 reviewers make very overlapping points about this extension being obtained from a straightforward combination of previous ideas. Furthermore, R3 is not convinced that the approach is well motivated, beyond “filling the gap” in the literature.

All reviewers also pointed out that the paper is very hard to read. The authors have improved the manuscript during the rebuttal, but the AC believes that the paper is still written in an unnecessarily complicated way.

Overall the AC believes that this paper needs some more work, specifically in (a) improving its presentation (b) providing more technical insights about the methods (as suggested by R2 and R3), which could be a means of boosting the novelty.